# Biased Stochastic First-Order Methods for Conditional Stochastic Optimization and Applications in Meta Learning

**Yifan Hu**[*]
UIUC
yifanhu3@illinois.edu

**Siqi Zhang**[*]
UIUC
siqiz4@illinois.edu

**Xin Chen**
UIUC
xinchen@illinois.edu

**Niao He**
UIUC & ETH Zurich
niao.he@inf.ethz.ch

## Abstract

Conditional stochastic optimization covers a variety of applications ranging from invariant learning and causal inference to meta-learning. However, constructing unbiased gradient estimators for such problems is challenging due to the composition structure. As an alternative, we propose a biased stochastic gradient descent (BSGD) algorithm and study the bias-variance tradeoff under different structural assumptions. We establish the sample complexities of BSGD for strongly convex, convex, and weakly convex objectives under smooth and non-smooth conditions. Our lower bound analysis shows that the sample complexities of BSGD cannot be improved for general convex objectives and nonconvex objectives except for smooth nonconvex objectives with Lipschitz continuous gradient estimator. For this special setting, we propose an accelerated algorithm called biased SpiderBoost (BSpiderBoost) that matches the lower bound complexity. We further conduct numerical experiments on invariant logistic regression and model-agnostic meta-learning to illustrate the performance of BSGD and BSpiderBoost.

## 1 Introduction

We study a class of optimization problems, called *conditional stochastic optimization (CSO)*:

$$\min_{x \in \mathcal{X}} F(x) := \mathbb{E}_\xi f_\xi(\mathbb{E}_{\eta|\xi} g_\eta(x, \xi)), \tag{1}$$

where $\mathcal{X} \subseteq \mathbb{R}^d$, $g_\eta(\cdot, \xi) : \mathbb{R}^d \to \mathbb{R}^k$ is a vector-valued function dependent on both random vectors $\xi$ and $\eta$, $f_\xi(\cdot) : \mathbb{R}^k \to \mathbb{R}$ depends on the random vector $\xi$, and the inner expectation is taken with respect to the conditional distribution of $\eta|\xi$. Throughout, we assume access to samples from the distribution $P(\xi)$ and the conditional distribution $P(\eta|\xi)$.

CSO includes the classical stochastic optimization as a special case when $g_\eta(x, \xi) = x$ but is much more general. It has been recently utilized to solve a variety of applications in machine learning, ranging from the policy evaluation and control in reinforcement learning [12, 13, 35], the optimal control in linearly-solvable Markov decision process [12], to instrumental variable regression in causal inference [41, 34].

One common challenge with these applications is that in the extreme case, a few or only one sample is available from the conditional distribution of $\eta|\xi$ for each given $\xi$. To deal with this limitation, a

---

[*]The first two authors have equal contribution.

primal-dual stochastic approximation algorithm was proposed to solve a min-max reformulation of CSO using the kernel embedding techniques [12]. However, this approach requires convexity of $f$ and linearity of $g$, which are not satisfied by general applications when neural networks are involved.

On the other hand, for many other applications, e.g., those arising from invariant learning and meta-learning, we do have access to multiple samples from the conditional distribution. Take the model-agnostic meta-learning (MAML) [19] as an example. MAML learns a meta-initialization parameter using metadata from similar learning tasks such that taking one or multiple gradient steps on a small training data would generalize well on a new task. It can be framed into the following CSO problem:

$$\min_{w} \ \mathbb{E}_{i \sim p, a \sim D^i_{query}} l_i \Big( \mathbb{E}_{b \sim D^i_{support}} \big( w - \alpha \nabla l_i(w, b) \big), a \Big), \tag{2}$$

where $p$ represents the distribution of different tasks, $D^i_{support}$ and $D^i_{query}$ correspond to support (training) data and query (testing) data of the task $i$, $l_i(\cdot, D^i)$ is the loss function on data $D^i$ from task $i$, and $\alpha$ is a fixed meta step size. Setting $\xi = (i, a)$ and $\eta = b$, (2) is clearly a special case of CSO for which multiple samples can be drawn from the conditional distribution of $P(\eta|\xi)$. Since the loss function generally involves neural networks, the resulting CSO is often nonconvex. Thus, the previous primal-dual algorithm and kernel embedding techniques developed in [12] no longer apply.

In this paper, we focus on the general CSO problem where multiple samples from the conditional distribution $P(\eta|\xi)$ are available, and the objective is not necessarily in the compositional form of a convex loss $f_\xi(\cdot)$ and a linear mapping $g_\eta(\cdot, \xi)$. Recently, Hu et al. [30] studied the generalization error bound and sample complexity of empirical risk minimization (ERM), a.k.a., sample average approximation (SAA) for general CSO:

$$\min_{x \in \mathcal{X}} \frac{1}{n} \sum_{i=1}^{n} f_{\xi_i} \Big( \frac{1}{m} \sum_{j=1}^{m} g_{\eta_{ij}}(x, \xi_i) \Big),$$

where $\{\xi_i\}_{i=1}^{n}$ are i.i.d. samples from $\mathbb{P}(\xi)$ and $\{\eta_{ij}\}_{j=1}^{m}$ are i.i.d. samples from $\mathbb{P}(\eta|\xi_i)$. They assumed that the global optimal solution to ERM can be computed without specifying how. Differently, here we aim at developing efficient stochastic gradient-based methods that directly solve the CSO problem (1) and find either a global optimal solution in the convex setting or a stationary point in the nonconvex setting, respectively.

Due to the composition structure of the CSO objective in (1), constructing unbiased gradient estimators is not possible in general. Instead, we leverage a mini-batch of conditional samples to construct the gradient estimator with controllable bias and propose a family of biased first-order methods, including (1) the biased stochastic gradient descent (BSGD) algorithm for general convex and nonconvex CSO objectives and (2) the biased SpiderBoost (BSpiderBoost) algorithm, designed for nonconvex smooth CSO objectives. Note that BSpiderBoost is inspired by the variance reduced method for nonconvex smooth stochastic optimization in [18, 44].

## 1.1 Our contributions

Our main results are summarized in Table 1. Our contributions are three-fold:

- We establish the first sample complexity results of BSGD and BSpiderBoost in the context of CSO. Since the bias of BSGD comes from estimating the conditional expectation rather than from a given stochastic oracle, the sample complexity closely depends on the smoothness conditions of the outer function, which is distinct from traditional SGD results. For convex problem, to achieve an $\epsilon$-optimal solution, the sample complexity of BSGD improves from $\mathcal{O}(\epsilon^{-4})$ to $\tilde{\mathcal{O}}(\epsilon^{-3})$ when either $f_\xi$ is smooth or $F$ is strongly convex and further improves to $\tilde{\mathcal{O}}(\epsilon^{-2})$ when both conditions hold, where $\tilde{\mathcal{O}}(\cdot)$ represents the bound with hidden logarithmic factors. For weakly convex CSO problems, BSGD requires a total sample complexity of $\mathcal{O}(\epsilon^{-8})$ to achieve an $\epsilon$-stationary point, and of $\mathcal{O}(\epsilon^{-6})$ when $f_\xi$ is smooth. If we further assume that both $f_\xi$ and $g_\eta$ are Lipschitz continuous and Lipschitz smooth, the biased gradient estimator is Lipschitz continuous, and then the sample complexity can be improved to $\mathcal{O}(\epsilon^{-5})$ by BSpiderBoost.
- We analyze the lower bounds on the minimax error of first-order algorithms using specific biased oracles for CSO objectives. With the upper bounds results, BSGD is optimal for strongly

Table 1: Sample Complexity for CSO

| Algorithm | Assumptions | | | | | | |
|---|---|---|---|---|---|---|---|
| $\hat{F}$ | *SC* | *SC* | *Convex* | *Convex* | *WC* | *WC* | *Smooth* |
| $f_\xi$ | *Smooth* | *Lipschitz* | *Smooth* | *Lipschitz* | *Smooth* | *Lipschitz* | *Smooth* |
| SAA [30]* | $\mathcal{O}(\epsilon^{-2})$ | $\mathcal{O}(\epsilon^{-3})$ | $\tilde{\mathcal{O}}(d\epsilon^{-3})$ | $\tilde{\mathcal{O}}(d\epsilon^{-4})$ | - | - | - |
| BSGD | $\tilde{\mathcal{O}}(\epsilon^{-2})$ | $\tilde{\mathcal{O}}(\epsilon^{-3})$ | $\mathcal{O}(\epsilon^{-3})$ | $\mathcal{O}(\epsilon^{-4})$ | $\mathcal{O}(\epsilon^{-6})$ | $\mathcal{O}(\epsilon^{-8})$ | $\mathcal{O}(\epsilon^{-6})$ |
| BSpiderBoost | - | - | - | - | - | - | $\mathcal{O}(\epsilon^{-5})$ |
| Lower Bound | $\mathcal{O}(\epsilon^{-2})$ | $\mathcal{O}(\epsilon^{-3})$ | $\mathcal{O}(\epsilon^{-3})$ | $\mathcal{O}(\epsilon^{-4})$ | $\mathcal{O}(\epsilon^{-6})$ | $\mathcal{O}(\epsilon^{-8})$ | $\mathcal{O}(\epsilon^{-5})$ |

**Goal**: find $\epsilon$-optimal solution for convex $F$ and $\epsilon$-stationary point for weakly convex $F$.
$\hat{F}$ is defined in (4). SC: strongly convex; WC: weakly convex; Lipschitz = Lipschitz continuous.
* SAA requires further solving the empirical risk minimization.

convex, convex, and weakly convex CSO objectives, and BSpiderBoost is optimal for the nonconvex smooth CSO problems under the additional oracle assumption that the gradient estimator returned by the oracle is Lipschitz continuous.

- When applied to MAML, BSGD converges to a stationary point under simple deterministic stepsize rules and appropriate inner mini-batch sizes. In contrast, the commonly used first-order MAML algorithm [19] ignores the Hessian information and is not guaranteed to converge even when a large inner mini-batch size is used. For smooth MAML, compared with the algorithm recently introduced in [17], BSGD without requiring stochastic stepsizes and mini-batches of outer samples at each iteration, thus is more practical. Leveraging the variance reduction technique, BSpiderBoost attains the best-known sample complexity for MAML, to our best knowledge. Numerically, we further demonstrate that BSGD and BSpiderBoost achieve superior performance for MAML.

## 1.2 Related work

**Nested expectation optimization (NEO)** [42, 43, 20, 45, 9, 47] deals with problems in the form of: $\min_{x \in \mathcal{X}} \mathbf{f} \circ \mathbf{g}(x) := \mathbb{E}_\xi \big[ f_\xi \big( \mathbb{E}_\eta [g_\eta(x)] \big) \big]$, where $\mathbf{f}(u) := \mathbb{E}_\xi [f_\xi(u)]$, $\mathbf{g}(x) := \mathbb{E}_\eta [g_\eta(x)]$. A key assumption is that even when $\xi$ and $\eta$ are dependent, there exists a *deterministic function* $\mathbf{g}(x)$, independent of $\xi$, which does not hold for general CSO. Hence their algorithms and analysis cannot extend to CSO. The best known sample complexity for smooth strongly convex NEO is $\mathcal{O}(\epsilon^{-1.25})$ [42]; and for nonconvex smooth NEO objective is $\mathcal{O}(\epsilon^{-4})$ [20] and $\tilde{\mathcal{O}}(\epsilon^{-3})$ [47] if $\xi$ and $\eta$ are independent using variance reduction technique.

**Nested expectation Estimation** Estimating nested expectations in the form of $\mathbb{E}[H(\mathbb{E}(\eta|\xi))]$ has been extensively studied in the statistics and simulation communities. [26, 24, 27] considered nested Monte Carlo estimator, when $H$ is a general non-linear function. [7, 22, 23] considered Multilevel Monte Carlo (MLMC) method [21] when $H$ has special structure. Note that this line of work purely focuses on estimation, whereas we deal with optimization, which is more challenging.

**Biased Gradient Methods** [29, 1, 32, 28, 8] analyzed the non-asymptotic convergence of general biased gradient methods. These papers assume that the bias in gradient estimator comes from certain black-box oracles or an additive non-zero mean noise. Differently in our problem, the bias directly comes from estimating the nested expectation and can be controlled by the sampling strategy.

**Notations** $\Pi_\mathcal{X}$ denotes the projection operator, i.e., $\Pi_\mathcal{X}(x) := \operatorname{argmin}_{z \in \mathcal{X}} \|z - x\|_2^2$. $\tilde{\mathcal{O}}(\cdot)$ represents the order hiding logarithmic factors. A function $f(\cdot) : \mathbb{R}^k \to \mathbb{R}$ is $L$-Lipschitz continuous on $\mathcal{X}$ if $|f(x) - f(y)| \leq L\|x - y\|_2$ holds for any $x, y \in \mathcal{X}$. A function $f(\cdot)$ is $S$-Lipschitz smooth on $\mathcal{X}$ if $f(x) - f(y) - \nabla f(y)^\top (x - y) \leq \frac{S}{2} \|x - y\|_2^2$ holds for any $x, y \in \mathcal{X}$. A function $f(\cdot)$ is $\mu$-convex on $\mathcal{X}$ if for any $x, y \in \mathcal{X}$, $f(x) - f(y) - \nabla f(y)^\top (x - y) \geq \frac{\mu}{2} \|x - y\|_2^2$. Note that $\mu > 0$, $\mu = 0$, and $\mu < 0$ correspond to $f$ being strongly convex, convex, and weakly convex, respectively. Lastly, we denote $x^* \in \operatorname{argmin}_{x \in \mathcal{X}} F(x)$ as an optimal solution to the problem of interest. For an abuse of notation, we use $\nabla$ to denote the Jacobian matrix, (sub)gradient vector, and derivative for simplicity.

---

**Algorithm 1** Biased Stochastic Gradient Descent (BSGD)

---

**Input:** Number of iterations $T$, inner mini-batch size $\{m_t\}_{t=1}^T$, initial point $x_1$, stepsize $\{\gamma_t\}_{t=1}^T$

1: **for** $t = 1$ to $T - 1$ **do**
2:    Sample $\xi_t$ from distribution $\mathbb{P}(\xi)$, and $m_t$ i.i.d samples $\{\eta_{tj}\}_{j=1}^{m_t}$ from distribution $P(\eta|\xi_t)$.
3:    Compute $\nabla \hat{F}(x_t; \xi_t, \{\eta_{tj}\}_{j=1}^{m_t})$ according to (3).
4:    Update
$$x_{t+1} = \Pi_{\mathcal{X}}\left(x_t - \gamma_t \nabla \hat{F}(x_t; \xi_t, \{\eta_{tj}\}_{j=1}^{m_t})\right).$$
5: **end for**

---

## 2   Biased Stochastic First-Order Methods

For simplicity, throughout, we assume that $\mathcal{X} \subseteq \mathbb{R}^d$ is closed and convex, and the random functions $f_\xi(\cdot)$ and $g_\eta(\cdot, \xi)$ are continuously differentiable. Based on the special composition structure of CSO and the chain rule, under mild conditions, the gradient of $F(x)$ in (1) is given by

$$\nabla F(x) = \mathbb{E}_\xi\left[(\mathbb{E}_{\eta|\xi} \nabla g_\eta(x, \xi))^\top \nabla f_\xi(\mathbb{E}_{\eta|\xi} g_\eta(x, \xi))\right].$$

Constructing an unbiased stochastic estimator of the gradient can be costly and even impossible. Instead, we consider a biased estimator of $\nabla F(x)$ using one sample $\xi$ and $m$ i.i.d. samples $\{\eta_j\}_{j=1}^m$ from the conditional distribution of $P(\eta|\xi)$ in the following form:

$$\nabla \hat{F}(x; \xi, \{\eta_j\}_{j=1}^m) := \left(\frac{1}{m} \sum\nolimits_{j=1}^m \nabla g_{\eta_j}(x, \xi)\right)^\top \nabla f_\xi\left(\frac{1}{m} \sum\nolimits_{j=1}^m g_{\eta_j}(x, \xi)\right). \tag{3}$$

Note that $\nabla \hat{F}(x; \xi, \{\eta_j\}_{j=1}^m)$ is the gradient of an empirical objective such that

$$\hat{F}(x; \xi, \{\eta_j\}_{j=1}^m) := f_\xi\left(\frac{1}{m} \sum\nolimits_{j=1}^m g_{\eta_j}(x, \xi)\right). \tag{4}$$

Based on this biased gradient estimator, we propose BSGD, which is formally described in Algorithm 1. When using fixed inner mini-batch sizes $m_t = m$, BSGD can be viewed as performing SGD updates on the surrogate objective $\mathbb{E}_{\{\xi, \{\eta_j\}_{j=1}^m\}} \hat{F}(x; \xi, \{\eta_j\}_{j=1}^m)$. Inspired by the recent success of variance-reduced methods for nonconvex stochastic optimization [37, 18, 44], we further introduce an accelerated algorithm BSpiderBoost, which is formally described in Algorithm 2. BSpiderBoost divides updates into "epoch": at the beginning of the epoch, it will initialize the gradient estimator with $N_1$ outer samples of $\xi$; then in later iterations in the epoch, the estimator will be updated with gradient information in current iteration generated with $N_2$ outer samples and the information from the last iteration. Compared to the classical SVRG method [31], this framework keeps utilizing the latest information for updates, which can generate more accurate gradient estimations.

Before presenting the main results, we make one observation that the bias of the function value estimator $\hat{F}$, induced by the composition structure, depends on the smoothness condition of the outer function $f_\xi$: for $L_f$-Lipschitz continuous $f_\xi$,

$$\mathbb{E}_{\xi, Y}[f_\xi(Y) - f_\xi(\mathbb{E}_{Y|\xi} Y)] \leq L_f \mathbb{E}_{\xi, Y|\xi} \|Y - \mathbb{E}_{Y|\xi} Y\|_2;$$

for $S_f$-Lipschitz smooth $f_\xi$,

$$\mathbb{E}_{\xi, Y}[f_\xi(Y) - f_\xi(\mathbb{E}_{Y|\xi} Y)] \leq \frac{S_f}{2} \mathbb{E}_{\xi, Y|\xi} \|Y - \mathbb{E}_{Y|\xi} Y\|_2^2,$$

where $Y$ is a random variable. To characterize the estimation error of $\hat{F}$, we make the following assumption.

**Assumption 2.1** *We assume that* $\sigma_g^2 := \sup_{\xi, x \in \mathcal{X}} \mathbb{E}_{\eta|\xi} \|g_\eta(x, \xi) - \mathbb{E}_{\eta|\xi} g_\eta(x, \xi)\|_2^2 < +\infty;$

Assumption 2.1 indicates that the random vector $g_\eta$ has bounded variance. Define

$$\Delta_f(m) = \begin{cases} L_f \sigma_g / \sqrt{m}, & \text{if } f_\xi \text{ is } L_f\text{-Lipschitz continuous,} \\ S_f \sigma_g^2 / 2m, & \text{if } f_\xi \text{ is } S_f\text{-Lipschitz smooth.} \end{cases} \tag{6}$$

The following lemmas characterize the estimation errors of $\hat{F}$ and $\nabla \hat{F}$.

**Algorithm 2** Biased SpiderBoost (BSpiderBoost)

---

**Input:** Number of iterations $T$, inner batch size $m$, stepsize $\gamma$, epoch length $q$, mini-batch sizes $N_1, N_2$,

1: **for** $t = 0$ to $T$ **do**
2:     **if** $\mod(t, q) = 0$ **then**
3:         Generate $N_1$ samples of $\{\xi_1, \cdots, \xi_{N_1}\}$
4:         Generate $m$ i.i.d samples $\{\eta_{ij}\}_{i=1}^m$ from $\mathbb{P}(\eta|\xi_i)$ for each $\xi_i \in \{\xi_1, \cdots, \xi_{N_1}\}$.
5:         Compute $v_t = \frac{1}{N_1} \sum_{i=1}^{N_1} \nabla \hat{F}(x_t; \xi_i, \{\eta_{ij}\}_{j=1}^m)$
6:     **else**
7:         Generate $N_2$ samples of $\{\xi_1, \cdots, \xi_{N_2}\}$
8:         Generate $m$ i.i.d samples $\{\eta_{ij}\}_{i=1}^m$ from $\mathbb{P}(\eta|\xi_i)$ for each $\xi_i \in \{\xi_1, \cdots, \xi_{N_2}\}$.
9:         Compute

$$v_t = \frac{1}{N_2} \sum_{i=1}^{N_2} \nabla \hat{F}(x_t; \xi_i, \{\eta_{ij}\}_{j=1}^m) - \frac{1}{N_2} \sum_{i=1}^{N_2} \nabla \hat{F}(x_{t-1}; \xi_i, \{\eta_{ij}\}_{j=1}^m) + v_{t-1} \qquad (5)$$

10:     **end if**
11:     Update $x_{t+1} = x_t - \gamma v_t$
12: **end for**
**Output:** $x_S$ which is uniformly randomly selected from $\{x_t\}_{t=1}^T$.

---

**Lemma 2.1 ([30])** *Under Assumption 2.1, for a sample $\xi$ and $m$ i.i.d. samples $\{\eta_j\}_{j=1}^m$ from the conditional distribution $P(\eta|\xi)$, and any $x \in \mathcal{X}$ that is independent of $\xi$ and $\{\eta_j\}_{j=1}^m$, we have*

$$\left| \mathbb{E}_{\{\xi, \{\eta_j\}_{j=1}^m\}} \hat{F}(x; \xi, \{\eta_j\}_{j=1}^m) - F(x) \right| \leq \Delta_f(m). \qquad (7)$$

This implies that, to control the estimation bias up to $\epsilon$, a number of $m = \mathcal{O}(\epsilon^{-2})$ samples is needed for Lipschitz continuous $f_\xi$ whereas $m = \mathcal{O}(\epsilon^{-1})$ is needed for Lipschitz smooth $f_\xi$.

**Lemma 2.2** *Under Assumption 2.1, if additionally assuming $f_\xi$ is $S_f$-Lipschitz smooth, $g_\eta$ is $L_g$-Lipschitz continuous, it holds that*

$$\| \mathbb{E} \nabla \hat{F}(x; \xi, \{\eta_j\}_{j=1}^m) - \nabla F(x) \|_2^2 \leq \frac{S_f^2 L_g^2 \sigma_g^2}{m}. \qquad (8)$$

## 3 Convergence Analysis

In this section, we provide the non-asymptotic convergence analysis of BSGD and BSpiderBoost. Our result illustrates how the smoothness condition of the outer function $f_\xi$ influences the inner sample complexity and the total sample complexity. This is quite different from the traditional SGD analysis for convex stochastic optimization, where the smoothness condition does not influence the complexity in terms of dependence on $\epsilon$ [6, 36]. Before showing the convergence, we impose an assumption on the convexity.

**Assumption 3.1** $\hat{F}(x; \xi, \{\eta_j\}_{j=1}^m)$ *is $\mu$-convex for any $m, \xi, \{\eta_j\}_{j=1}^m$.*

Strong convexity, namely when $\mu > 0$, can often be achieved by adding $\ell_2$-regularization to convex objectives. Convexity, namely when $\mu = 0$, holds when (i) $f_\xi$ is convex and $g_\eta$ is linear; (ii) $f_\xi$ and $g_\eta$ are convex and $f_\xi$ is non-decreasing. Weak convexity, namely when $\mu < 0$, holds when (i) $F$ is Lipschitz smooth, which holds if (i) both $f_\xi$ and $g_\eta$ are Lipschitz continuous and smooth and (ii) $f_\xi$ is convex and $g_\eta$ is Lipschitz smooth [14]. Note that weak convexity is commonly used in nonconvex optimization literature [39, 10, 14, 48]. Beyond weak convexity, little is known on the complexity of first-order algorithms except for some special functions, e.g., TAME functions [15] and difference of convex functions [38]. Lastly, we point out that weak convexity is satisfied by various objectives used in machine learning, e.g., MAML discussed in this paper. Note that under mild conditions, Assumption 3.1 implies that $F(x)$ is $\mu$-convex.

**Global convergence of BSGD for strongly convex objectives.** We have the following result:

**Theorem 3.1** *Under Assumption 2.1 and Assumption 3.1 with $\mu > 0$, if $\hat{F}$ is $S_F$-Lipschitz smooth, there exists a constant Set $\gamma_t = \frac{1}{\mu(t+c)}$ with $c = \max\{4S_F^2/\mu^2 - 1, 0\}$, the output $\hat{x}_T = \frac{1}{T}\sum_{t=1}^T x_t$ of BSGD satisfies:*

$$\mathbb{E}[F(\hat{x}_T) - F(x^*)] \leq \frac{2\mathbb{E}\|\nabla\hat{F}(x^*)\|_2^2(\log(T)+1) + S_F^2/4\|x_1 - x^*\|_2^2}{T\mu} + \frac{4}{T}\sum_{t=1}^T \Delta_f(m_t). \quad (9)$$

Hence, to achieve $\epsilon$-optimality, the number of iterations $T$ should be at least $\tilde{\mathcal{O}}(\epsilon^{-1})$, which aligns with the performance of SGD for strongly convex objectives [40]. For strongly-convex CSO with Lipschitz continuous $f_\xi$, recall the definition of $\Delta_f(m_t)$ in (6), using a fixed mini-batch size $m_t = \mathcal{O}(\epsilon^{-2})$ or time varying batch sizes $m_t = \mathcal{O}(t^2)$ would be sufficient to obtain $\epsilon$-optimality. For strongly-convex CSO with Lipschitz smooth $f_\xi$, it suffices to set $m_t = \mathcal{O}(\epsilon^{-1})$ or $m_t = \mathcal{O}(t)$. Respectively, the total sample complexities are $\tilde{\mathcal{O}}(\epsilon^{-3})$ and $\tilde{\mathcal{O}}(\epsilon^{-2})$ under these two settings.

**Global convergence of BSGD for convex objectives** We make an additional assumption about the second moment of the gradient estimator.

**Assumption 3.2** *There exists $M > 0$ such that $\mathbb{E}\big[\|\nabla\hat{F}(x; \xi, \{\eta_j\}_{j=1}^m)\|_2^2 \mid x\big] \leq M^2$ for any $x$.*

Note that Assumption 3.2 is common in the literature for analyzing SGD when the objective is non-strongly-convex or nonsmooth, e.g., when $f_\xi$ and $g_\eta$ are $L_f$- and $L_g$- Lipschitz continuous, $M = L_f L_g$. See e.g., [36, 6, 14].

**Theorem 3.2** *Under Assumptions 2.1 Assumption 3.1 with $\mu = 0$, and Assumption 3.2, with stepsizes $\gamma_t = c/\sqrt{T}$ for a positive constant c, the output $\hat{x}_T = \frac{1}{T}\sum_{t=1}^T x_t$ of BSGD satisfies*

$$\mathbb{E}[F(\hat{x}_T) - F(x^*)] \leq \frac{M^2c^2 + \|x_1 - x^*\|_2^2}{2c\sqrt{T}} + \frac{2}{T}\sum_{t=1}^T \Delta_f(m_t).$$

Comparing to Theorem 3.1, without strong convexity condition, the iteration complexity increases from $\mathcal{O}(\epsilon^{-1})$ to $\mathcal{O}(\epsilon^{-2})$. The total sample complexity for convex CSO is $\mathcal{O}(\epsilon^{-4})$ for Lipschitz continuous $f_\xi$ with $m_t = \mathcal{O}(\epsilon^{-2})$ or $m_t = \mathcal{O}(t)$ and $\mathcal{O}(\epsilon^{-3})$ for Lipschitz smooth $f_\xi$ with $m_t = \mathcal{O}(\epsilon^{-1})$ or $m_t = \mathcal{O}(\sqrt{t})$.

**Stationary convergence of BSGD for general nonconvex objectives** When $F(x)$ is nonconvex and possibly nonsmooth, we first introduce the notion of convergence using Moreau envelope $F_\lambda(x)$ ($\lambda > 0$) of function $F(x)$ and its corresponding minimizer:

$$F_\lambda(x) := \min_{z \in \mathcal{X}}\left\{F(z) + \frac{1}{2\lambda}\|z - x\|_2^2\right\}, \ \text{prox}_{\lambda F}(x) := \operatorname*{argmin}_{z \in \mathcal{X}}\left\{F(z) + \frac{1}{2\lambda}\|z - x\|_2^2\right\}.$$

Based on Moreau envelope, we define the gradient mapping: $\mathcal{G}_{\lambda F}(x) := \frac{1}{\lambda}\|\text{prox}_{\lambda F}(x) - x\|_2$. We say $x$ is an $\epsilon$-stationary point of $F$ if $\mathbb{E}[\mathcal{G}_{\lambda F}(x)] \leq \epsilon$. This convergence criterion is commonly used in nonconvex optimization literature [4, 16]. We have the following result.

**Theorem 3.3** *Under Assumption 2.1, Assumption 3.1 with $\mu < 0$, and Assumption 3.2, with stepsizes $\gamma_t = c/\sqrt{T}$ for a positive constant c, the output, $\hat{x}_R$, selected uniformly randomly from $\{x_1, \cdots, x_T\}$, satisfies*

$$\mathbb{E}\big[\mathcal{G}_{\frac{1}{2|\mu|}F}^2(\hat{x}_R)\big] \leq \frac{2F_{1/(2|\mu|)}(x_1) - 2F(x^*) + 2|\mu|M^2c^2}{c\sqrt{T}} + \frac{8|\mu|}{T}\sum_{t=1}^T \Delta_f(m_t).$$

To the best of our knowledge, this is the first non-asymptotic convergence guarantee for CSO in the nonconvex setting. Specifically, for nonconvex CSO with Lipschitz continuous $f_\xi$, setting batch sizes $m_t = \mathcal{O}(\epsilon^{-4})$ or $m_t = \mathcal{O}(t)$ yields a total sample complexity of $\mathcal{O}(\epsilon^{-8})$; for nonconvex CSO with Lipschitz smooth $f_\xi$, using $m_t = \mathcal{O}(\epsilon^{-2})$ or $m_t = \mathcal{O}(\sqrt{t})$ achieves the total sample complexity of $\mathcal{O}(\epsilon^{-6})$. Note that $\mathcal{O}(\epsilon^{-6})$ sample complexity also holds when the CSO objective is additionally smooth. The analysis only requires little modification and is omitted.

**Stationary convergence of BSpiderBoost for nonconvex smooth objectives**   We now analyze the stationary convergence of BSpiderBoost for CSO problem with $\mathcal{X} = \mathbb{R}^d$ and Lipschitz smooth $F(x)$. We say $x$ is an $\epsilon$-stationary point of $F$ if $\mathbb{E}\|\nabla F(x)\|_2 \leq \epsilon$. Before proceeding, we make the following assumption:

**Assumption 3.3** $f_\xi(\cdot)$ is $L_f$-Lipschitz continuous and $S_f$-Lipschitz smooth for any $\xi$. $g_\eta(\cdot, \xi)$ are $L_g$-Lipschitz continuous and $S_g$-Lipschitz smooth for any $\xi$ and $\eta$.

This assumption ensures that $F$ and $\hat{F}$ are $S_F$-Lipschitz smooth with $S_F = S_g L_f + S_f L_g^2$.

**Theorem 3.4 (Convergence of BSpiderBoost)** *Under Assumptions 2.1 and 3.3, $\mathcal{X} = \mathbb{R}^d$, and $\Delta :=$ $F(x_0) - F^* < \infty$, consider the following setup: $T = \lceil 8\Delta\beta^{-1}\epsilon^{-2}\rceil$, $q = \lfloor\sqrt{N_1}\rfloor$, $N_2 = \lceil 2\sqrt{N_1}\rceil$, $\gamma_t \equiv \gamma = \frac{1}{2S_F}$, and*

$$N_1 = \left\lceil \left(3 + \frac{3}{2\beta S_F} + \frac{3}{16\beta S_F}\right)\frac{4L_f^2 L_g^2}{\epsilon^2} \right\rceil, \quad m = \left\lceil \left(3 + \frac{3}{2\beta S_F} + \frac{3}{16\beta S_F}\right)\frac{4L_g^2 S_f^2 \sigma_g^2}{\epsilon^2} \right\rceil,$$

*where*

$$\beta := \frac{\gamma(1 - S_F\gamma)}{2} - \frac{\gamma^3 S_F^2 q}{N_2} \geq \frac{1}{16S_F} > 0.$$

*The output of BSpiderBoost, $x_S$, which is randomly drawn from $\{x_1, ..., x_T\}$, attains $\mathbb{E}\|\nabla F(x_S)\|_2 \leq \epsilon$. Correspondingly, the sample complexity of BSpiderBoost is $\mathcal{O}(\epsilon^{-5})$.*

**Remark 3.1** *Recall that the objective of MAML (2) is a special case of CSO. If $\nabla l_i$ and $\nabla^2 l_i$ is Lipschitz continuous, then the objective is smooth and the outer function is smooth. Thus BSGD converges to an $\epsilon$-stationary point of (2) with sample complexity $\mathcal{O}(\epsilon^{-6})$ and BSpiderBoost converges with sample complexity $\mathcal{O}(\epsilon^{-5})$.*

# 4   Lower Bounds for Conditional Stochastic Optimization

In this section, we show that the sample complexity of BSGD for (strongly) convex and general nonconvex CSO objectives cannot be improved without further assumptions. The sample complexity achieved by BSpiderBoost also cannot be improved for nonconvex smooth CSO objectives. The analysis uses the well-known oracle model, which consists of three components: a function class of interest $\mathcal{F}$, an algorithm class $\mathcal{A}$, and an oracle class $\Phi$. Specifically, we consider the biased stochastic first-order oracle class for CSO denoted as $\Phi_m$, where $m$ is the fixed number of conditional samples used. We also consider $\Phi_m^c$, a subset of $\Phi_m$ such that any oracle $\phi$

**Definition 4.1 (Biased first-order oracle for CSO)** *For a query at point $x$ of CSO objective $F$ given by an algorithm, an oracle $\phi \in \Phi_m$ with a parameter $\sigma^2$ takes a sample $\zeta$ from its associated distribution $P(\zeta)$, and returns to the algorithm $\phi(x, F) = (h(x, \zeta), G(x, \zeta))$ such that*

$$\mathbb{E}h(x, \zeta) = \mathbb{E}\hat{F}(x; \xi, \{\eta_j\}_{j=1}^m); \quad \mathbb{E}G(x, \zeta) = \nabla\mathbb{E}h(x, \zeta); \quad \mathbb{E}\|G(x, \zeta) - \mathbb{E}G(x, \zeta)\|_2^2 \leq \sigma^2.$$

*In addition, we define the oracle class $\Phi_m^c$ such that $\Phi_m^c \subset \Phi_m$ and any oracle $\phi \in \Phi_m^c$ will return to the algorithm a Lipschitz continuous gradient estimator $G(\cdot, \zeta)$ for any $\zeta$.*

**Function class** we use $\mathcal{F}_{\text{CSO}}$ to denote the CSO function class of interest. Specifically, we use $\mathcal{F}_{\text{CSO}}$ with superscript $+, 0, -$ to denote strongly convex, convex, nonconvex function class.

**Randomized Algorithm** A randomized algorithm class $\mathcal{A}$ contains algorithms $A$ such that $A$ maps the oracle output and a random seed $r$ to the next query point $x_{t+1}^A(\phi) = A(r, G(x_t^A(\phi), \zeta))$.

**Updating Procedure** Suppose an algorithm $A \in \mathcal{A}$ is applied to minimize a function $F \in \mathcal{F}$ using oracle $\phi \in \Phi$. The updating procedure is such that at iteration $0$, the algorithm starts with some initialization point $x_0$. At iteration $t$, the algorithm $A$ queries the oracle $\phi$ about the information about $F$ on $x_t^A(\phi)$. The oracle $\phi$ will return some (noisy) information $\phi(x_t^A(\phi), F)$ back to the algorithm. Then the algorithm would base on all previous information returned by the oracle to generate the next query point $x_{t+1}^A(\phi)$.

For a fixed number of iteration $T$, we define the minimax error as:

$$\Delta_T^*(\mathcal{A}, \mathcal{F}, \Phi) := \inf_{A \in \mathcal{A}} \sup_{\phi \in \Phi} \sup_{F \in \mathcal{F}} \Delta_T(A, F, \phi) := \mathbb{E}F(x_T^A(\phi)) - \min_{x \in \mathcal{X}} F(x);$$

$$\Delta_T^{*g}(\mathcal{A}, \mathcal{F}, \Phi) := \inf_{A \in \mathcal{A}} \sup_{\phi \in \Phi} \sup_{F \in \mathcal{F}} \Delta_T^g(A, F, \phi) := \mathbb{E}\|\nabla F(x_T^A(\phi))\|_2^2, \tag{10}$$

where the expectation is taken with respect to the randomness in algorithm $A$ and oracle $\phi$. $\Delta_T^*$ is used to capture the global optimality for convex function classes and $\Delta_T^{*g}$ is used to capture the stationarity of the output for nonconvex function classes. If $\Delta_T^* \geq \epsilon$, it implies that for any algorithm $A$, there exists a 'hard' function $F$ and an oracle $\phi$ such that the expected optimization error incurred by $A$ is at least $\epsilon$.

**Theorem 4.1** *For CSO problem, the minimax error satisfies that*

*(i) when $f_\xi$ is Lipschitz continuous,*

$$\Delta_T^*(\mathcal{A}, \mathcal{F}_{\mathrm{CSO}}^+, \Phi_m) \geq \mathcal{O}(m^{-1/2} + \sigma^2 T^{-1}); \quad \Delta_T^*(\mathcal{A}, \mathcal{F}_{\mathrm{CSO}}^0, \Phi_m) \geq \mathcal{O}(m^{-1/2} + \sigma T^{-1/2});$$
$$\Delta_T^{*g}(\mathcal{A}, \mathcal{F}_{\mathrm{CSO}}^-, \Phi_m) \geq \mathcal{O}(m^{-1/2} + \sigma T^{-1/2}).$$

*(ii) when $f_\xi$ is Lipschitz smooth,*

$$\Delta_T^*(\mathcal{A}, \mathcal{F}_{\mathrm{CSO}}^+, \Phi_m) \geq \mathcal{O}(m^{-1} + \sigma^2 T^{-1}); \quad \Delta_T^*(\mathcal{A}, \mathcal{F}_{\mathrm{CSO}}^0, \Phi_m) \geq \mathcal{O}(m^{-1} + \sigma T^{-1/2});$$
$$\Delta_T^{*g}(\mathcal{A}, \mathcal{F}_{\mathrm{CSO}}^-, \Phi_m) \geq \mathcal{O}(m^{-1} + \sigma T^{-1/2}).$$

*(iii) when the gradient estimator is Lipschitz continuous and $f_\xi$ is Lipschitz smooth,*

$$\Delta_T^{*g}(\mathcal{A}, \mathcal{F}_{\mathrm{CSO}}^-, \Phi_m^c) \geq \mathcal{O}(m^{-1} + \sigma T^{-2/3}).$$

Together with Theorems 3.1, 3.2, and 3.3, Theorem 4.1 demonstrates that the sample complexity of BSGD cannot be further improved for strongly convex, convex and weakly convex CSO problems without any additional Lipschitz continuity assumption on the gradient estimator. Similarly, the sample complexity of BSpiderBoost cannot be improved for the nonconvex smooth CSO problems.

## 5 Numerical Experiments

In this section, we illustrate the performance of the proposed algorithms on the invariant logistic regression and MAML. The detailed experiment setup, results, and platform information are deferred to Appendix D.

**Invariant Logistic Regression**  Invariant learning has wide applications in training robust classifiers [33, 2]. We consider the invariant logistic regression problem:

$$\min_w \mathbb{E}_{\xi=(a,b)}\big[\log(1 + \exp(-b\mathbb{E}_{\eta|\xi}[\eta^T w]))\big], \tag{11}$$

where $a \in \mathbb{R}^d$ is the random feature vector, $b \in \{\pm 1\}$ is the corresponding label and $\eta$ is a random perturbed observation of the feature $a$. Let $\sigma_1^2, \sigma_2^2$ denote the variances of $a$ and $\eta|a$, respectively. We observe that for a given budget of total samples, BSGD outperforms SAA and converges even when a small inner batch size is used as shown in Table 2. Detailed results are in Table 4 in Appendix D.1.

Table 2: Comparison of BSGD and SAA

| $\sigma_2^2/\sigma_1^2$ | **BSGD** | | | **SAA** | | |
|---|---|---|---|---|---|---|
| | $m$ | *Mean* | *Dev* | $m$ | *Mean* | *Dev* |
| 1 | 5 | 1.77e-04 | 4.70e-05 | 100 | 5.56e-04 | 2.81e-04 |
| 10 | 5 | 3.26e-04 | 1.15e-04 | 464 | 2.14e-03 | 8.45e-04 |
| 100 | 50 | 1.50e-03 | 6.97e-04 | 1000 | 1.12e-02 | 6.42e-04 |

Figure 1 summarizes the performance of BSGD with different inner batch sizes and under different noise ratios for a given total number of samples. When the noise ratio $\sigma_2^2/\sigma_1^2$ increases, more inner samples are needed to achieve the same performance, as suggested by the theory.

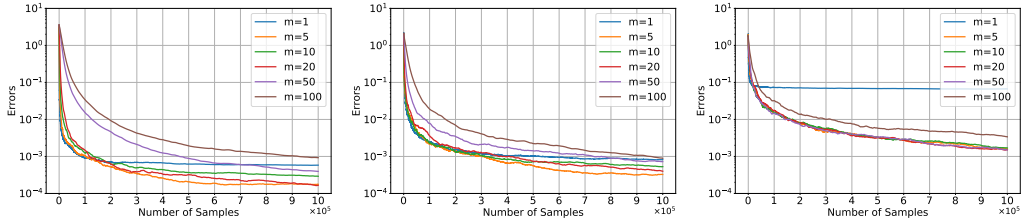

Figure 1: BSGD for invariant Logistic regression (a) $\sigma_2^2/\sigma_1^2 = 1$, (b) $\sigma_2^2/\sigma_1^2 = 10$, (c) $\sigma_2^2/\sigma_1^2 = 100$.

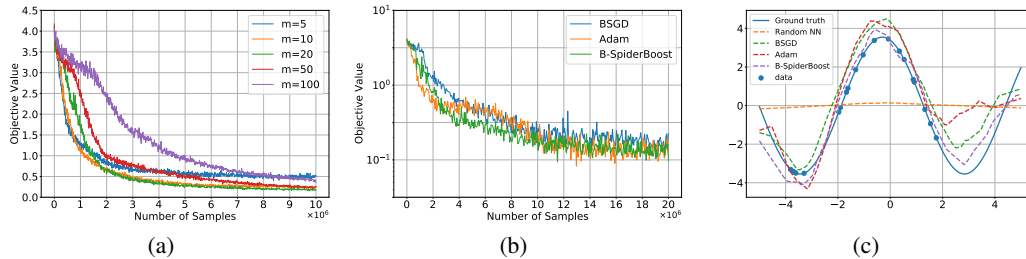

Figure 2: (a) Convergences of BSGD under differnt inner batch size. (b) Convergences of BSGD, Adam and BSpiderBoost. (c) Recovered sine-wave signals on an unseen task.

**Model-Agnostic Meta-Learning (MAML)** We consider the widely used sine-wave few-shot learning. The goal is to find a good initialization model parameter such that the network could recover a new unseen sine wave using only a few available data points.

The sine wave is of the form $y = a\sin(x + b)$ where $(a, b)$ are drawn from a task distribution. Recall the MAML formulation in (2). In this experiments, we set $\alpha = 0.01$, $l_i(w, D^i) = (y^i - h_i(w, x^i))^2$, where $D^i = (x^i, y^i)$ is the data for the $i$-th task and $h_i$ is a neural network consisting of 2 hidden layers with 40 nodes and ReLU activation function between each layers. We evaluate the MAML objective via empirical objective obtained by empirical risk minimization.

Figure 2(a) demonstrates a tradeoff between the inner batch size $m$ and the number of iterations for BSGD. Figure 2(b) compares the convergence performance of BSGD, Adam, and BSpiderBoost with the best tuned inner batch sizes. Here Adam refers to a variant of BSGD that performs Adam updates using the biased gradient estimator we constructed. Figure 2(c) shows the recovered signal after a one-step update on the unseen task with only 20 samples using the initialization model parameters obtained by all three algorithms in the meta training step. Random NN refers to the recovered signal using the neural network with random initialization.

Table 3 summarizes the average loss and running time (in CPU minutes) over 10 trials of each algorithm (under their best inner batch sizes). Although the widely used first-order MAML (FO-MAML) [19] requires the least running time, its performance is worse than BSGD. When $m = 50$, FO-MAML does not converge (Figure 3 in Appendix). BSGD requires a smaller batch size to achieve its best performance, which is more practical when a task only has a small number of samples.

Table 3: Comparison of the average loss and average running time

| | $\alpha = 0.01, Q = 10^7$ | | | | | |
| $m$ | **BSGD** | | **FO-MAML** | | **Adam** | |
| | *Mean* | *CPU* | *Mean* | *CPU* | *Mean* | *CPU* |
| 10 | 2.12e-01 | 71.57 | 2.52e-01 | 41.45 | 8.16e-01 | 86.54 |
| 20 | 2.04e-01 | 35.63 | 2.50e+00 | 20.60 | 3.99e-01 | 43.42 |
| 50 | 2.17e-01 | 14.63 | 3.98e+00 | 8.64 | 2.77e-01 | 17.62 |

To summarize, BSpiderBoost achieves the best recovery result but is much harder to tune in practice. In terms of convergence, BSpiderBoost is marginally better than BSGD on this sine-wave task. A possible reason might be that the objective function in our example is not necessarily smooth due to the ReLU activation. Ramdon NN could fail the MAML task when there is a limited amount of samples. More details are available in Appendix D.2.

## Broader Impact

This paper is purely theoretical and has no immediate ethical or societal consequences.

## Acknowledgments and Disclosure of Funding

We thank all reviewers and the area chair for the detailed feedback. Funding for this work was provided by NSF CRII under the award number CCF-1755829.

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
