[Supplementary Material]

# Appendix

## A  Proof of Lemma 2.2

**Proof:**  Denote $\hat{g}(x,\xi) := \frac{1}{m}\sum_{j=1}^{m} g_{\eta_j}(x,\xi)$. Note that

$$\|\mathbb{E}\nabla\hat{F}(x;\xi,\{\eta_j\}_{j=1}^m) - \nabla F(x)\|_2^2$$

$$\leq \left\|\mathbb{E}_\xi \mathbb{E}_{\{\eta_j\}_{j=1}^m|\xi}\left(\nabla\hat{g}(x,\xi)\right)^\top \nabla f_\xi\left(\hat{g}(x,\xi)\right) - \mathbb{E}_\xi(\mathbb{E}_{\eta|\xi}\nabla g_\eta(x,\xi))^\top \nabla f_\xi(\mathbb{E}_{\eta|\xi}g_\eta(x,\xi))\right\|_2^2$$

$$\leq \left\|\mathbb{E}_\xi \mathbb{E}_{\{\eta_j\}_{j=1}^m|\xi}\left(\nabla\hat{g}(x,\xi)\right)^\top \nabla f_\xi\left(\hat{g}(x,\xi)\right) - \mathbb{E}_\xi \mathbb{E}_{\{\eta_j\}_{j=1}^m|\xi}\left(\nabla\hat{g}(x,\xi)\right)^\top \nabla f_\xi\left(\mathbb{E}_{\eta|\xi}g_\eta(x,\xi)\right)\right\|_2^2$$

$$+ \left\|\mathbb{E}_\xi \mathbb{E}_{\{\eta_j\}_{j=1}^m|\xi}\left(\nabla\hat{g}(x,\xi)\right)^\top \nabla f_\xi\left(\mathbb{E}_{\eta|\xi}g_\eta(x,\xi)\right) - \mathbb{E}_\xi(\mathbb{E}_{\eta|\xi}\nabla g_\eta(x,\xi))^\top \nabla f_\xi(\mathbb{E}_{\eta|\xi}g_\eta(x,\xi))\right\|_2^2$$

$$= \left\|\mathbb{E}_\xi \mathbb{E}_{\{\eta_j\}_{j=1}^m|\xi}\left(\nabla\hat{g}(x,\xi)\right)^\top \nabla f_\xi\left(\hat{g}(x,\xi)\right) - \mathbb{E}_\xi \mathbb{E}_{\{\eta_j\}_{j=1}^m|\xi}\left(\nabla\hat{g}(x,\xi)\right)^\top \nabla f_\xi\left(\mathbb{E}_{\eta|\xi}g_\eta(x,\xi)\right)\right\|_2^2$$

$$\leq \mathbb{E}_\xi \mathbb{E}_{\{\eta_j\}_{j=1}^m|\xi}\|\nabla\hat{g}(x,\xi)\|_2^2 \|\nabla f_\xi\left(\hat{g}(x,\xi)\right) - \nabla f_\xi\left(\mathbb{E}_{\eta|\xi}g_\eta(x,\xi)\right)\|_2^2$$

$$\leq L_g^2 S_f^2 \mathbb{E}_\xi \mathbb{E}_{\{\eta_j\}_{j=1}^m|\xi}\|\hat{g}(x,\xi) - \mathbb{E}_{\eta|\xi}g_\eta(x,\xi)\|_2^2$$

$$\leq \frac{L_g^2 S_f^2 \sigma_g^2}{m}.$$

The equality holds as

$$\mathbb{E}_\xi \mathbb{E}_{\{\eta_j\}_{j=1}^m|\xi}\left(\nabla\hat{g}(x,\xi)\right)^\top \nabla f_\xi\left(\mathbb{E}_{\eta|\xi}g_\eta(x,\xi)\right) - \mathbb{E}_\xi(\mathbb{E}_{\eta|\xi}\nabla g_\eta(x,\xi))^\top \nabla f_\xi(\mathbb{E}_{\eta|\xi}g_\eta(x,\xi))$$

$$= \mathbb{E}_\xi \mathbb{E}_{\{\eta_j\}_{j=1}^m|\xi}(\nabla\hat{g}(x,\xi) - \mathbb{E}_{\eta|\xi}\nabla g_\eta(x,\xi))^\top \nabla f_\xi(\mathbb{E}_{\eta|\xi}g_\eta(x,\xi))$$

$$= 0.$$

$\square$

## B  Convergence Analysis

In this section, we present the proof of Theorems 3.1, 3.2, and 3.3. Based on these theorems, we demonstrate the sample complexity of BSGD with strongly convex, convex, and weakly convex objectives.

First, we present the proof framework for strongly convex and convex objectives. Recall BSGD in Algorithm 1 , at iteration $t$, BSGD first generates sample $\xi_t$ from the distribution of $\xi$ and $m$ samples $\{\eta_{tj}\}_{j=1}^{m_t}$ from the conditional distribution of $\eta|\xi_t$. We define the following auxiliary functions to facilitate our analysis:

$$p(x,\xi_t) := f_{\xi_t}(\mathbb{E}_{\eta|\xi_t}g_\eta(x,\xi_t)); \quad \hat{p}(x,\xi_t) := f_{\xi_t}\left(\frac{1}{m_t}\sum_{j=1}^{m_t}g_{\eta_{tj}}(x,\xi_t)\right).$$

Note that $\hat{F}(x;\xi_t,\{\eta_{tj}\}_{j=1}^{m_t}) = \hat{p}(x,\xi_t)$. The biased gradient estimator used in BSGD is $\nabla\hat{p}(x,\xi_t)$. Denote $x^* \in \operatorname{argmin}_{x\in\mathcal{X}}F(x)$, $A_t = \frac{1}{2}\|x_t - x^*\|_2^2$, $a_t = \mathbb{E}A_t$. Since $\Pi_\mathcal{X}(x^*) = x^*$ and the projection operator is non-expansive, we have

$$
\begin{aligned}
A_{t+1} &= \frac{1}{2}\|x_{t+1} - x^*\|_2^2 \\
&= \frac{1}{2}\|\Pi_\mathcal{X}(x_t - \gamma_t\nabla_x\hat{p}(x_t,\xi_t)) - \Pi_\mathcal{X}(x^*)\|_2^2 \\
&\leq \frac{1}{2}\|x_t - x^* - \gamma_t\nabla_x\hat{p}(x_t,\xi_t)\|_2^2 \\
&= A_t + \frac{1}{2}\gamma_t^2\|\nabla_x\hat{p}(x_t,\xi_t)\|_2^2 - \gamma_t\nabla_x\hat{p}(x_t,\xi_t)^\top(x_t - x^*).
\end{aligned}
$$

(12)

Dividing $\gamma_t$ on both sides and taking expectation over $\{\xi_t, \{\eta_{tj}\}_{j=1}^{m_t}\}$, it holds

$$\mathbb{E}\nabla_x\hat{p}(x_t, \xi_t)^\top(x_t - x^*) \leq \frac{a_t - a_{t+1}}{\gamma_t} + \frac{1}{2}\gamma_t\mathbb{E}\|\nabla_x\hat{p}(x_t, \xi_t)\|_2^2. \tag{13}$$

By Assumption 3.1, we have

$$-\nabla_x\hat{p}(x_t, \xi_t)^\top(x_t - x^*) \leq \hat{p}(x^*, \xi_t) - \hat{p}(x_t, \xi_t) - \frac{\mu}{2}\|x_t - x^*\|_2^2$$

$$= \underbrace{\hat{p}(x^*, \xi_t) - p(x^*, \xi_t)}_{:=\zeta_{t1}} + \underbrace{p(x^*, \xi_t) - p(x_t, \xi_t)}_{:=\zeta_{t2}} + \underbrace{p(x_t, \xi_t) - \hat{p}(x_t, \xi_t)}_{:=\zeta_{t3}} - \frac{\mu}{2}\|x_t - x^*\|_2^2. \tag{14}$$

Taking expectation over $\{\xi_t, \{\eta_{tj}\}_{j=1}^{m_t}\}$ on both sides, by the definition of $p(x, \xi)$, it holds $\mathbb{E}_{\xi_t}[\zeta_{t2} \mid x_t] = \mathbb{E}_{\xi_t}[F(x^*) - F(x_t) \mid x_t]$, then

$$-\mathbb{E}\nabla_x\hat{p}(x_t, \xi_t)^\top(x_t - x^*) \leq \mathbb{E}\zeta_{t1} + \mathbb{E}\zeta_{t3} + \mathbb{E}[F(x^*) - F(x_t)] - \mu a_t. \tag{15}$$

Since $x^*$ and $x_t$ are independent of $\{\xi_t, \{\eta_{tj}\}_{j=1}^m\}$, by Lemma 2.1, we upper bound $\mathbb{E}\zeta_{t1}$ and $\mathbb{E}\zeta_{t3}$ using $\Delta_f(m_t)$:

$$|\mathbb{E}\zeta_{t1}| \leq \Delta_f(m_t), \quad |\mathbb{E}\zeta_{t3}| \leq \Delta_f(m_t).$$

Summing up (13) and (15), we obtain

$$\mathbb{E}[F(x_t) - F(x^*)] \leq 2\Delta_f(m_t) - \mu a_t + \frac{a_t - a_{t+1}}{\gamma_t} + \frac{1}{2}\gamma_t\mathbb{E}\|\nabla_x\hat{p}(x_t, \xi_t)\|_2^2. \tag{16}$$

By convexity of $F$ and the definition of $\hat{x}_T = \frac{1}{T}\sum_{t=1}^T x_t$, we have

$$\mathbb{E}[F(\hat{x}_T) - F(x^*)] = \mathbb{E}\left[F\left(\frac{1}{T}\sum_{t=1}^T x_t\right) - F(x^*)\right] \leq \frac{1}{T}\sum_{t=1}^T \mathbb{E}[F(x_t) - F(x^*)], \tag{17}$$

We then prove the convergence of BSGD for strongly convex and convex objectives based on (16).

## B.1  Global Convergence of BSGD for Strongly Convex Objectives

We prove Theorem 3.1, the strongly convex case for which Assumption 3.1 holds with $\mu > 0$.

**Proof:**  Since $\hat{F}$ is $S_F$-Lipschitz smooth and $\mu$-strongly convex, we have

$$\mathbb{E}\|\nabla\hat{F}(x)\|_2^2 \leq 2\mathbb{E}\|\nabla\hat{F}(x) - \nabla\hat{F}(x^*)\|_2^2 + 2\mathbb{E}\|\nabla\hat{F}(x^*)\|_2^2$$
$$\leq 2S^2\|x - x^*\|_2^2 + 2\mathbb{E}\|\nabla\hat{F}(x^*)\|_2^2 \tag{18}$$
$$\leq 4S^2/\mu(F(x) - F(x^*)) + 2\mathbb{E}\|\nabla\hat{F}(x^*)\|_2^2.$$

It implies that

$$\mathbb{E}[F(x_t) - F(x^*)] \leq 2\Delta_f(m_t) - \mu a_t + \frac{a_t - a_{t+1}}{\gamma_t} + \frac{1}{2}\gamma_t(4S_F^2/\mu(F(x) - F(x^*)) + 2\mathbb{E}\|\nabla\hat{F}(x^*)\|_2^2). \tag{19}$$

Therefore, we have for $\gamma_t \leq \frac{\mu}{4S_F^2}$,

$$\mathbb{E}F(x_t) - F(x^*) \leq \frac{1}{1 - \gamma_t S_F^2/\mu}\left(2\Delta_f(m_t) - \mu a_t + \frac{a_t - a_{t+1}}{\gamma_t} + \gamma_t\mathbb{E}\|\nabla\hat{F}(x^*)\|_2^2\right)$$

$$\leq 2\left(2\Delta_f(m_t) - \mu a_t + \frac{a_t - a_{t+1}}{\gamma_t} + \gamma_t\mathbb{E}\|\nabla\hat{F}(x^*)\|_2^2\right). \tag{20}$$

Summing up (20) from $t = 1$ to $T$,

$$\frac{1}{T}\sum_{t=1}^T \mathbb{E}[F(x_t) - F(x^*)]$$

$$\leq \frac{2}{T}\sum_{t=1}^T \left[2\Delta_f(m_t) - \mu a_t + \frac{a_t - a_{t+1}}{\gamma_t} + \gamma_t\mathbb{E}\|\nabla\hat{F}(x^*)\|_2^2\right] \tag{21}$$

$$\leq \frac{2}{T}\sum_{t=1}^T \left[2\Delta_f(m_t) + \gamma_t\mathbb{E}\|\nabla\hat{F}(x^*)\|_2^2\right] + \frac{2}{T}\sum_{t=2}^T a_t\left(\frac{1}{\gamma_t} - \frac{1}{\gamma_{t-1}} - \mu\right) + \frac{2}{T}a_1\left(\frac{1}{\gamma_1} - \mu\right).$$

Set $\gamma_t = \frac{1}{\mu(t+c)}$ and $c = \max\{4S_F^2/\mu^2 - 1, 0\}$. It makes sure that $\gamma_t \le \gamma_1 \le \frac{\mu}{4S_F^2}$. Since $1/\gamma_1 - \mu \le \mu(4S_F^2/\mu^2 - 1)$, with inequality (17), it holds

$$\mathbb{E}[F(\hat{x}_T) - F(x^*)] \le \frac{4}{T}\sum_{t=1}^{T}\Delta_f(m_t) + \frac{1}{T}\sum_{t=1}^{T}\frac{2\mathbb{E}\|\nabla\hat{F}(x^*)\|_2^2}{\mu(t+c)} + \frac{S_F^2}{4\mu T}\|x_1 - x^*\|_2^2.$$

By the fact that $\sum_{t=1}^{T}\frac{1}{t+c} \le \sum_{t=1}^{T}\frac{1}{t} \le \log(T) + 1$, it holds

$$\mathbb{E}[F(\hat{x}_T) - F(x^*)] \le \frac{4}{T}\sum_{t=1}^{T}\Delta_f(m_t) + \frac{2\mathbb{E}\|\nabla\hat{F}(x^*)\|_2^2(\log(T)+1) + S_F^2/4\|x_1 - x^*\|_2^2}{T\mu}.$$

$\square$

We demonstrate the sample complexity using the following corollary.

**Corollary B.1** *To achieve an $\epsilon$-optimal solution, the total sample complexity of BSGD in the strongly convex case is $\tilde{\mathcal{O}}(\epsilon^{-3})$ for objectives with Lipschitz continuous $f_\xi$ and $\tilde{\mathcal{O}}(\epsilon^{-2})$ for objectives with Lipschitz smooth $f_\xi$.*

It implies that the smoothness of the outer function makes a difference in the total sample complexity of BSGD when solving CSO. It is worth pointing out that the sample complexity of BSGD matches with that of ERM (SAA) for strongly convex objectives established in Hu et al. [30]. We now prove Corollary B.1.

**Proof:** For fixed mini-batch size, to guarantee that $\mathbb{E}[F(\hat{x}_T) - F(x^*)] \le \epsilon$, setting $T = \tilde{\mathcal{O}}(\epsilon^{-1})$ and picking $m = \mathcal{O}(\epsilon^{-2})$ for objectives with Lipschitz continuous outer function $f_\xi$ and $m = \mathcal{O}(\epsilon^{-1})$ for objectives with Lipschitz smooth outer function $f_\xi$ are sufficient to guarantee that $\hat{x}_T$ is an $\epsilon$-optimal solution to the (1).

As for time-varying mini-batch sizes, letting $m_t = t^2$ for Lipschitz continuous $f_\xi$. Since $\sum_{t=1}^{T}\frac{1}{t} \le \log(T) + 1$, it holds

$$\frac{1}{T}\sum_{t=1}^{T}\Delta_f(m_t) = \frac{1}{T}\sum_{t=1}^{T}\frac{L_f\sigma_g}{t} \le \frac{L_f\sigma_g(\log(T)+1)}{T} \le \mathcal{O}(\epsilon).$$

As a result, setting $T = \tilde{\mathcal{O}}(\epsilon^{-1})$; the total sample complexity is $\sum_{t=1}^{T}(m_t + 1) = \mathcal{O}(T^3) = \tilde{\mathcal{O}}(\epsilon^{-3})$.

Set $m_t = t$ for Lipschitz smooth $f_\xi$. Since $\sum_{t=1}^{T}\frac{1}{t} \le \log(T) + 1$, it holds

$$\frac{1}{T}\sum_{t=1}^{T}\Delta_f(m_t) \le \frac{1}{T}\sum_{t=1}^{T}\frac{S_f\sigma_g^2}{2m_t} \le \frac{S_f\sigma_g^2(\log(T)+1)}{2T} \le \mathcal{O}(\epsilon).$$

Setting $T = \tilde{\mathcal{O}}(\epsilon^{-1})$, the total sample complexity is $\sum_{t=1}^{T}(m_t + 1) = \mathcal{O}(T^2) = \tilde{\mathcal{O}}(\epsilon^{-2})$ for objectives with Lipschitz smooth $f_\xi$. $\square$

## B.2 Global Convergence of BSGD for Convex Objectives

We prove Theorem 3.2, the convex case for which Assumption 3.1 holds with $\mu = 0$.

**Proof:** Recall that

$$\mathbb{E}[F(\hat{x}_T) - F(x^*)] \le \frac{1}{T}\mathbb{E}\sum_{t=1}^{T}[F(x_t) - F(x^*)].$$

Since $\beta = 0$ and $\mu = 0$, summing up (16) from $t = 1$ to $T$,

$$\frac{1}{T}\sum_{t=1}^{T}\mathbb{E}[F(x_t) - F(x^*)] \le \frac{1}{T}\sum_{t=1}^{T}\left[2\Delta_f(m_t) + \frac{a_t - a_{t+1}}{\gamma_t} + \frac{1}{2}\gamma_t\mathbb{E}\|\nabla_x\hat{p}(x_t, \xi_t)\|_2^2\right]$$

$$\le \frac{1}{T}\sum_{t=1}^{T}\left[2\Delta_f(m_t) + \frac{1}{2}\gamma_t M^2\right] + \frac{1}{T}\sum_{t=2}^{T}a_t\left(\frac{1}{\gamma_t} - \frac{1}{\gamma_{t-1}}\right) + \frac{1}{\gamma_1 T}a_1.$$

Plugging constant stepsizes $\gamma_t = \gamma$ and $a_1 = \|x_1 - x^*\|_2^2/2$, we have

$$\mathbb{E}[F(\hat{x}_T) - F(x^*)] \leq \frac{2}{T}\sum_{t=1}^{T}\Delta_f(m_t) + \frac{1}{2}\gamma M^2 + \frac{\|x_1 - x^*\|_2^2}{2T\gamma}.$$

Setting $\gamma = \frac{c}{\sqrt{T}}$, we have the desired result

$$\mathbb{E}[F(\hat{x}_T) - F(x^*)] \leq \frac{2}{T}\sum_{t=1}^{T}\Delta_f(m_t) + \frac{M^2c^2 + \|x_1 - x^*\|_2^2}{2c\sqrt{T}}. \tag{22}$$

$\square$

Comparing to Nemirovski et al. [36] and Hazan et al. [25], (22) has an extra term $\frac{2}{T}\sum_{t=1}^{T}\Delta_f(m_t)$ that represents the average estimation bias of the function value estimator $\hat{p}(x, \xi_t)$ over $F(x)$.

**Corollary B.2** *Under the same assumptions as Theorem 3.2, to achieve an $\epsilon$-optimal solution, the total sample complexity required by BSGD is $\mathcal{O}(\epsilon^{-4})$ for convex CSO objectives with Lipschitz continuous $f_\xi$ and $\mathcal{O}(\epsilon^{-3})$ for convex CSO objectives with Lipschitz smooth $f_\xi$.*

The sample complexity is achieved for either fixed mini-batch size $m_t = m$ or the time-varying mini-batch sizes $m_t = t$ for Lipschitz continuous $f_\xi$ or $m_t = \lceil\sqrt{t}\rceil$ for Lipschitz smooth $f_\xi$.

**Proof:** Let $T = \mathcal{O}(\epsilon^{-2})$. For fixed inner batch sizes $m_t = m$, the selection of $m$ is obvious by definition of $\Delta_f(m_t)$.

For time-varying batch sizes, when $f_\xi$ is Lipschitz continuous, let $m_t = t$. Invoking $\sum_{t=1}^{T} 1/\sqrt{t} \leq 2\sqrt{T}$, we have,

$$\mathbb{E}[F(\hat{x}_T) - F(x^*)] \leq \frac{2}{T}\sum_{t=1}^{T}\frac{L_f\sigma_g}{\sqrt{t}} + \frac{M^2c^2 + \|x_1 - x^*\|_2^2}{2c\sqrt{T}} \leq \frac{4L_f\sigma_g}{\sqrt{T}} + \frac{M^2c^2 + \|x_1 - x^*\|_2^2}{2c\sqrt{T}} \leq \epsilon.$$

The sample complexity is $\sum_{t=1}^{T}(t+1) = \mathcal{O}(T^2) = \mathcal{O}(\epsilon^{-4})$.

When $f_\xi$ is Lipschitz smooth, letting $m_t = \lceil\sqrt{t}\rceil$, we have

$$\mathbb{E}[F(\hat{x}_T) - F(x^*)] \leq \frac{2}{T}\sum_{t=1}^{T}\frac{S_f\sigma_g^2}{2\sqrt{t}} + \frac{M^2c^2 + \|x_1 - x^*\|_2^2}{2c\sqrt{T}} \leq \frac{2S_f\sigma_g^2}{\sqrt{T}} + \frac{M^2c^2 + \|x_1 - x^*\|_2^2}{2c\sqrt{T}} \leq \epsilon.$$

The sample complexity is $\sum_{t=1}^{T}(\sqrt{t}+1) = \mathcal{O}(T^{3/2}) = \mathcal{O}(\epsilon^{-3})$. $\square$

### B.3 Stationarity Convergence of BSGD for Weakly Convex Objectives

We prove Theorem 3.3. In this case, Assumption 3.1 with $\mu < 0$ implies that $F(x)$ is $|\mu|$-weakly convex. For simplicity, we denote $x_t' := \text{prox}_{\lambda F}(x_t)$. $\lambda$ is specified later in the proof.

**Proof:** By the definition of Moreau envelope, we have for any $\hat{\mu} > |\mu|$,

$$F_{1/\hat{\mu}}(x_{t+1}) \leq F(x_t') + \frac{\hat{\mu}}{2}\|x_t' - x_{t+1}\|^2$$

$$\leq F(x_t') + \hat{\mu}\gamma_t\nabla\hat{p}(x_t, \xi_t)^\top(x_t' - x_{t+1}) + \frac{\hat{\mu}}{2}\|x_t' - x_t\|^2 - \frac{\hat{\mu}}{2}\|x_{t+1} - x_t\|^2$$

$$= F_{1/\hat{\mu}}(x_t) + \hat{\mu}\gamma_t\nabla\hat{p}(x_t, \xi_t)^\top(x_t' - x_{t+1}) - \frac{\hat{\mu}}{2}\|x_{t+1} - x_t\|^2$$

$$= F_{1/\hat{\mu}}(x_t) + \hat{\mu}\gamma_t\nabla\hat{p}(x_t, \xi_t)^\top(x_t' - x_t) + \hat{\mu}\gamma_t\nabla\hat{p}(x_t, \xi_t)^\top(x_t - x_{t+1}) - \frac{\hat{\mu}}{2}\|x_{t+1} - x_t\|^2$$

$$= F_{1/\hat{\mu}}(x_t) + \hat{\mu}\gamma_t\nabla\hat{p}(x_t, \xi_t)^\top(x_t' - x_t) + \frac{\hat{\mu}\gamma_t^2\|\nabla\hat{p}(x_t, \xi_t)\|_2^2}{2},$$

$$\tag{23}$$

where the second inequality comes from the triangle inequality, and the last equality comes from plugging in $x_{t+1} - x_t$. By weak convexity of $\widehat{p}(\cdot)$, we have

$$
\begin{aligned}
&\nabla \widehat{p}(x_t, \xi_t)^\top (x_t' - x_t) \\
&\leq \widehat{p}(x_t', \xi_t) - \widehat{p}(x_t, \xi_t) + \frac{|\mu|}{2} ||x_t' - x_t||^2 \\
&\leq \underbrace{\widehat{p}(x_t', \xi_t) - p(x_t', \xi_t)}_{:=\zeta_{t1}} + \underbrace{p(x_t', \xi_t) - p(x_t, \xi_t)}_{:=\zeta_{t2}} + \underbrace{p(x_t, \xi_t) - \widehat{p}(x_t, \xi_t)}_{:=\zeta_{t3}} + \frac{|\mu|}{2} ||x_t' - x_t||^2.
\end{aligned}
\tag{24}
$$

By definition, $\mathbb{E}_{\xi_t} p(x, \xi_t) = F(x)$. Invoking Lemma 2.1, $|\mathbb{E}\zeta_{t1}| \leq \Delta_f(m_t)$, $|\mathbb{E}\zeta_{t3}| \leq \Delta_f(m_t)$. Combining (23) and (24), taking expectation over $\{\xi_t, \{\eta_{tj}\}_{j=1}^{m_t}\}$ on both sides, and using the fact that $\mathbb{E}||\nabla\widehat{p}(x,\xi)||_2^2 \leq M^2$, we have

$$
F_{1/\hat{\mu}}(x_{t+1}) - F_{1/\hat{\mu}}(x_t) \leq \hat{\mu}\gamma_t(2\Delta_f(m_t) + F(x_t') - F(x_t) + \frac{|\mu|}{2}||x_t' - x_t||^2) + \frac{\hat{\mu}\gamma_t^2 M^2}{2}.
$$

Dividing $\hat{\mu}$ on both sides, rearranging and summing up from $t = 1$ to $T$, we have

$$
\begin{aligned}
&\sum_{t=1}^T \gamma_t \Big( F(x_t) - F(x_t') - \frac{|\mu|}{2}||x_t' - x_t||^2 \Big) \\
&\leq \frac{1}{\hat{\mu}} \Big( F_{1/\hat{\mu}}(x_1) - F_{1/\hat{\mu}}(x_{T+1}) + \frac{\hat{\mu}M^2 \sum_{t=1}^T \gamma_t^2}{2} \Big) + 2 \sum_{t=1}^T \gamma_t \Delta_f(m_t).
\end{aligned}
\tag{25}
$$

We divide $\sum_{t=1}^T \gamma_t$ on both sides of the inequality above. Recall the definition of the output of the algorithm $\hat{x}_R$. Since $\gamma_t / \sum_{t=1}^T \gamma_t = 1/T$ due to the constant stepsize and $\hat{x}_R$ is selected from $\{x_1, ..., x_T\}$ with equal probability, we have

$$
\begin{aligned}
&\mathbb{E}\Big[ F(\hat{x}_R) - F(\hat{x}_R') - \frac{|\mu|}{2}||\hat{x}_R' - \hat{x}_R||^2 \Big] \\
&\leq \frac{F_{1/\hat{\mu}}(x_1) - F_{1/\hat{\mu}}(x_{T+1}) + \frac{1}{2}\hat{\mu}M^2 \sum_{t=1}^T \gamma_t^2 + 2\hat{\mu}\sum_{t=1}^T \gamma_t \Delta_f(m_t)}{\hat{\mu}\sum_{t=1}^T \gamma_t}.
\end{aligned}
\tag{26}
$$

Noticing that $F(z) + \frac{\hat{\mu}}{2}||z - x||^2$ is $(\hat{\mu} - |\mu|)$-strongly convex if $\hat{\mu} > |\mu|$. Setting $\lambda = 1/\hat{\mu}$, we have

$$
\begin{aligned}
&F(x_t) - F(x_t') - \frac{|\mu|}{2}||x_t' - x_t||^2 \\
&= (F(x_t) + \frac{\hat{\mu}}{2}||x_t - x_t||^2) - (F(x_t') + \frac{\hat{\mu}}{2}||x_t' - x_t||^2) + \frac{\hat{\mu} - |\mu|}{2}||x_t' - x_t||^2 \\
&\geq (\hat{\mu} - |\mu|)||x_t' - x_t||^2 = \frac{\hat{\mu} - |\mu|}{\hat{\mu}^2} \mathcal{G}_{1/\hat{\mu}F}^2(x_t),
\end{aligned}
$$

where the last inequality uses the strong convexity of $F(z) + \frac{\hat{\mu}}{2}||z - x||_2^2$. Recall that $\mathcal{G}_{\lambda F}(x) := \frac{1}{\lambda}||\text{prox}_{\lambda F}(x) - x||_2$. Combining with (26), we obtain

$$
\mathbb{E}\big[\mathcal{G}_{1/\hat{\mu}F}^2(\hat{x}_R)\big] \leq \frac{\hat{\mu}}{\hat{\mu} - |\mu|} \frac{F_{1/\hat{\mu}}(x_1) - F_{1/\hat{\mu}}(x_{T+1}) + \frac{1}{2}\hat{\mu}M^2 \sum_{t=1}^T \gamma_t^2 + 2\hat{\mu}\sum_{t=1}^T \gamma_t \Delta_f(m_t)}{\sum_{t=1}^T \gamma_t}.
\tag{27}
$$

Plugging $\gamma_t = c/\sqrt{T}$ and $\hat{\mu} = 2|\mu|$ into the expression above, we have

$$
\begin{aligned}
&\mathbb{E}\big[\mathcal{G}_{1/(2|\mu|)F}^2(\hat{x}_R)\big] \\
&\leq 2\frac{F_{1/(2|\mu|)}(x_1) - F_{1/(2|\mu|)}(x_{T+1}) + |\mu|M^2 T\frac{c^2}{T} + 4|\mu|\frac{c}{\sqrt{T}}\sum_{t=1}^T \Delta_f(m_t)}{T \cdot \frac{c}{\sqrt{T}}} \\
&= 2\frac{F_{1/(2|\mu|)}(x_1) - F_{1/(2|\mu|)}(x_{T+1}) + |\mu|M^2 c^2}{c\sqrt{T}} + \frac{8|\mu|\sum_{t=1}^T \Delta_f(m_t)}{T}.
\end{aligned}
$$

By the fact that $F_{1/(2|\mu|)}(x_{T+1}) \geq \inf_{x \in \mathcal{X}} F(x)$, we conclude the proof. $\square$

**Corollary B.3** *Under the same assumptions as Theorem 3.3, to achieve an $\epsilon$-stationary point, the total sample complexity required by BSGD is at most $\mathcal{O}(\epsilon^{-8})$. If further assuming Lipschitz smooth $f_\xi$, the sample complexity is at most $\mathcal{O}(\epsilon^{-6})$.*

The proof and batch size selection are the same as the convex case. We also provide a convergence guarantee using decaying stepsizes.

**Corollary B.4** *(Decaying Stepsizes) Let $T \geq 3$, inner batch size $m_t \equiv m$, and stepsize $\gamma_t = c/\sqrt{t}$ $(t = 1, \cdots, T)$ with $c > 0$. If the output $\hat{x}_R$ is chosen from $\{x_1, \ldots, x_T\}$ with $P(\hat{x}_R = x_i) = \gamma_i / \sum_{t=1}^{T} \gamma_t$, $(i = 1, \cdots, T)$, we have,*

$$\mathbb{E}[\mathcal{G}^2_{1/(2|\mu|)F}(\hat{x}_R)] \leq \frac{2F_{1/(2|\mu|)}(x_1) - 2\min_{x \in \mathcal{X}} F(x) + 4|\mu|c^2 M^2 \ln T}{c\sqrt{T}} + 8|\mu|\Delta_f(m).$$

**Proof:**   Note that the argument till (26) still applies.

Plugging $\gamma_t = \frac{c}{\sqrt{t}}$ into (27), we have

$$\mathbb{E}\big[\mathcal{G}^2_{1/(2|\mu|)F}(\hat{x}_R)\big]$$
$$\leq 2 \cdot \left( \frac{F_{1/(2|\mu|)}(x_1) - F_{1/(2|\mu|)}(x_{T+1}) + |\mu|M^2 \sum_{t=1}^{T}\gamma_t^2}{\sum_{t=1}^{T}\gamma_t} + \frac{4|\mu|\sum_{t=1}^{T}\gamma_t\Delta_f(m)}{\sum_{t=1}^{T}\gamma_t} \right)$$
$$\leq \frac{2F_{1/(2|\mu|)}(x_1) - 2F_{1/(2|\mu|)}(x_{T+1}) + 2|\mu|M^2\sum_{t=1}^{T}\gamma_t^2}{\sum_{t=1}^{T}\gamma_t} + 8|\mu|\Delta_f(m).$$

Note that for $T \geq 3$

$$\sum_{t=1}^{T} t^{-\frac{1}{2}} \geq \int_{1}^{T+1} t^{-\frac{1}{2}}dt = 2(\sqrt{T+1} - 1) \geq \sqrt{T};$$
$$\sum_{t=1}^{T} t^{-1} \leq 1 + \int_{2}^{T} t^{-1}dt = 1 + \ln T \leq 2\ln T.$$

We conclude the proof.   $\square$

### B.4   Stationary Convergence of BSpiderBoost for Nonconvex Smooth Objectives

To analyze the convergence, define the following auxiliary functions:

$$\widehat{F}_m(x) := \mathbb{E}_\xi \mathbb{E}_{\{\eta_i|\xi\}_{i=1}^m} f_\xi\left( \frac{1}{m}\sum_{i=1}^{m} g_{\eta_i}(x, \xi) \right),$$

where $\{\eta_i\}_{i=1}^m$ are i.i.d samples from the conditional distribution $\mathbb{P}(\eta|\xi)$. We summarize the properties of $F$ and $\hat{F}_m$ as follows:

**Proposition B.1** *Under Assumptions 2.1, 3.3, it holds that*

(a). $F(x)$ and $\hat{F}_m(x)$ are $S_F$-Lipschitz smooth where $S_F = S_g L_f + S_f L_g^2$.

(b). $\mathbb{E}_\xi \|\nabla f_\xi(y) - \nabla \mathbb{E}f_\xi(y)\|_2^2 \leq L_f^2$, $\mathbb{E}_{\xi,\eta}\|\nabla g_\eta(x,\xi) - \nabla \mathbb{E}g_\eta(x,\xi)\|_2^2 \leq L_g^2$, $\|\nabla \hat{F}_m(x)\|_2^2 \leq L_f^2 L_g^2$.

(c). By Lemma 2.1, $\|F(x) - \widehat{F}_m(x)\|_2^2 \leq \frac{L_f^2 \sigma_g^2}{m}$

(d). By Lemma 2.2, $\|\nabla F(x) - \nabla \widehat{F}_m(x)\|_2^2 \leq \frac{L_g^2 S_f^2 \sigma_g^2}{m}$.

Note that there are other conditions under which these properties would hold, for instance, a natural sufficient condition to ensure the smoothness of $F(\cdot)$ and $\hat{F}_m(\cdot)$ is when $f$ or $g$ is linear, i.e. $S_f$ or $S_g$ equals to zero.

**Proof:** Denote $g(x,\xi) = \mathbb{E}_{\eta|\xi} g_\eta(x,\xi)$, $\hat{g}(x,\xi) := \frac{1}{m}\sum_{j=1}^m g_{\eta_j}(x,\xi)$. By definition,

$$\nabla F(x) = \nabla \mathbb{E}_\xi\Big[f_\xi\big(g(x,\xi)\big)\Big] = \mathbb{E}_\xi\Big[\nabla\big(f_\xi(g(x,\xi))\big)\Big] = \mathbb{E}_\xi\Big[\nabla f_\xi\big(g(x,\xi)\big)\cdot\nabla g(x,\xi)\Big].$$

Note that for each fixed $\xi$, we have

$$\|\nabla\big(f_\xi(g(x,\xi))\big) - \nabla\big(f_\xi(g(y,\xi))\big)\|_2 = \|\nabla f_\xi(g(x,\xi))\cdot\nabla g(x,\xi) - \nabla f_\xi(g(y,\xi))\cdot\nabla g(y,\xi)\|_2$$
$$= \|\nabla f_\xi(g(x,\xi))\cdot\nabla g(x,\xi) - \nabla f_\xi(g(x,\xi))\cdot\nabla g(y,\xi) + \nabla f_\xi(g(x,\xi))\cdot\nabla g(y,\xi) - \nabla f_\xi(g(y,\xi))\cdot\nabla g(y,\xi)\|_2$$
$$\leq \|\nabla f_\xi(g(x,\xi))\cdot\big(\nabla g(x,\xi) - \nabla g(y,\xi)\big)\|_2 + \|\big(\nabla f_\xi(g(x,\xi)) - \nabla f_\xi(g(y,\xi))\big)\cdot\nabla g(y,\xi)\|_2$$
$$\leq L_f\|\nabla g(x,\xi) - \nabla g(y,\xi)\|_2 + S_f\|\big(g(x,\xi) - g(y,\xi)\big)\cdot\nabla g(y,\xi)\|_2$$
$$\leq (S_g L_f + S_f L_g^2)\|x - y\|_2,$$

where the last inequality comes from Lipschitz continuity and Lipschitz smoothness of $g_\eta(\cdot,\xi)$. Similarly,

$$\nabla\Big(f_\xi\big(\hat{g}(x,\xi)\big)\Big) = \nabla f_\xi\big(\hat{g}(x,\xi)\big)^\top \nabla \hat{g}(x,\xi).$$

$$\|\nabla\Big(f_\xi\big(\hat{g}(x,\xi)\big)\Big) - \nabla\Big(f_\xi(\hat{g}(y,\xi))\Big)\|_2$$
$$= \|\nabla\hat{g}(x,\xi)^\top\nabla f_\xi(\hat{g}(x,\xi)) - \nabla\hat{g}(y,\xi)^\top\nabla f_\xi(\hat{g}(y,\xi))\|_2$$
$$= \|\nabla\hat{g}(x,\xi)^\top\nabla f_\xi(\hat{g}(x,\xi)) - \nabla\hat{g}(y,\xi)^\top\nabla f_\xi(\hat{g}(x,\xi)) + \nabla\hat{g}(y,\xi)^\top\nabla f_\xi(\hat{g}(x,\xi)) - \nabla\hat{g}(y,\xi)^\top\nabla f_\xi(\hat{g}(y,\xi))\|_2$$
$$\leq \|\big(\nabla\hat{g}(x,\xi) - \nabla\hat{g}(y,\xi)\big)^\top\nabla f_\xi(\hat{g}(x,\xi))\|_2 + \|\nabla\hat{g}(y,\xi)^\top\big(\nabla f_\xi(\hat{g}(x,\xi)) - \nabla f_\xi(\hat{g}(y,\xi))\big)\|_2$$
$$\leq L_f\|\nabla\hat{g}(x,\xi) - \nabla\hat{g}(y,\xi)\|_2 + S_f\|\hat{g}(x,\xi) - \hat{g}(y,\xi)\|_2\cdot\|\nabla g(y,\xi)\|_2$$
$$\leq (S_g L_f + S_f L_g^2)\|x - y\|_2.$$

It concludes the proof of Proposition B.1(a).

As for Proposition B.1(b), note that for any random variables $X$, we have $\mathbb{E}\|X - EX\|_2^2 \leq \mathbb{E}\|X\|_2^2$. It implies that

$$\mathbb{E}_\xi\|\nabla f_\xi(y) - \nabla\mathbb{E} f_\xi(y)\|_2^2 \leq \mathbb{E}_\xi\|\nabla f_\xi(y)\|_2^2 \leq L_f^2,$$
$$\mathbb{E}_{\xi,\eta}\|\nabla g_\eta(x,\xi) - \nabla\mathbb{E} g_\eta(x,\xi)\|_2^2 \leq \mathbb{E}_{\xi,\eta}\|\nabla g_\eta(x,\xi)\|_2^2 \leq L_g^2. \tag{28}$$

It further holds that

$$\mathbb{E}_\xi\mathbb{E}_{\{\eta_i|\xi\}_{i=1}^m}\Big\|\nabla\Big(f_\xi\big(\hat{g}(x,\xi)\big)\Big) - \nabla\widehat{F}_m(x)\Big\|_2^2$$
$$\leq \mathbb{E}_\xi\mathbb{E}_{\{\eta_i|\xi\}_{i=1}^m}\Big\|\nabla\Big(f_\xi\big(\hat{g}(x,\xi)\big)\Big)\Big\|_2^2$$
$$= \mathbb{E}_\xi\mathbb{E}_{\{\eta_i|\xi\}_{i=1}^m}\Big\|\nabla f_\xi\big(\hat{g}(x,\xi)\big)^\top\nabla\big(\hat{g}(x,\xi)\big)\Big\|_2^2 \tag{29}$$
$$\leq \mathbb{E}_\xi\mathbb{E}_{\{\eta_i|\xi\}_{i=1}^m}\Big\|\nabla f_\xi\big(\hat{g}(x,\xi)\big)\Big\|_2^2\cdot\Big\|\frac{1}{m}\sum_{i=1}^m\nabla g_{\eta_i}(x,\xi)\Big\|_2^2$$
$$\leq L_f^2\mathbb{E}_\xi\mathbb{E}_{\{\eta_i|\xi\}_{i=1}^m}\Big[\frac{1}{m}\sum_{i=1}^m\Big\|\nabla g_{\eta_i}(x,\xi)\Big\|_2^2\Big] \leq L_f^2 L_g^2.$$

Proposition B.1(c) and (d) are direct implications of Lemma 2.1 and Lemma 2.2, respectively.

$\square$

Recall the gradient estimator $v_t$ of BSpiderBoost

$$v_t = \begin{cases} \nabla F_m^{N_2}(x_t) - \nabla F_m^{N_2}(x_{t-1}) + v_{t-1} & (n_t - 1)q + 1 \leq t \leq n_t q - 1, \\ \nabla F_m^{N_1}(x_t) & t = (n_t - 1)q, \end{cases} \tag{30}$$

where $n_t = \lceil t/q \rceil$. Different from a key step in the analysis of SPIDER related literature [18, 44], the sequence $\{v_t - \nabla F(x_t)\}$ in our work is not a martingale sequence because $\nabla F_m^{N_2}(x_t) - \nabla F_m^{N_2}(x_{t-1})$

is not an unbiased estimator of $\nabla F(x_t) - \nabla F(x_{t-1})$. As a result, the error would accumulate during iterations. To fix the issue, we consider the following sequence in our analysis

$$\{v_t - \nabla \widehat{F}_m(x_t)\}_t. \tag{31}$$

which is a martingale. The key lemma of SPIDER is formulated as follows:

**Lemma B.1 ([18], Lemma 1)** *Denote $n_t = \lceil t/q \rceil$, then for all $(n_t - 1)q + 1 \le t \le n_t q - 1$, we have*

$$\mathbb{E}\|v_t - \nabla \widehat{F}_m(x_t)\|_2^2 \le \frac{S_F^2}{N_2}\mathbb{E}\|x_t - x_{t-1}\|_2^2 + \mathbb{E}\|v_{t-1} - \nabla \widehat{F}_m(x_{t-1})\|_2^2, \tag{32}$$

*and by telescoping and Proposition B.1 we have*

$$\mathbb{E}\|v_t - \nabla \widehat{F}_m(x_t)\|_2^2 \le \mathbb{E}\|v_{(n_t-1)q} - \nabla \widehat{F}_m(x_{(n_t-1)q})\|_2^2 + \sum_{i=(n_t-1)q}^{t-1} \frac{S_F^2}{N_2}\mathbb{E}\|x_{i+1} - x_i\|_2^2$$

$$\le \frac{L_f^2 L_g^2}{N_1} + \sum_{i=(n_t-1)q}^{t-1} \frac{S_F^2}{N_2}\mathbb{E}\|x_{i+1} - x_i\|_2^2. \tag{33}$$

**Proposition B.2** *For each iteration, we have*

$$F(x_{t+1}) \le F(x_t) - \frac{\gamma(1 - S_F\gamma)}{2}\|v_t\|_2^2 + \gamma\Big(\|\nabla F(x_t) - \nabla \widehat{F}_m(x_t)\|_2^2 + \|\nabla \widehat{F}_m(x_t) - v_t\|_2^2\Big). \tag{34}$$

**Proof:** By smoothness of $F(\cdot)$, we have

$$F(x_{t+1}) \le F(x_t) + \nabla F(x_t)^\top (x_{t+1} - x_t) + \frac{S_F}{2}\|x_{t+1} - x_t\|^2$$

$$= F(x_t) - \gamma\nabla F(x_t)^\top v_t + \frac{S_F\gamma^2}{2}\|v_t\|^2$$

$$= F(x_t) - \gamma v_t^\top (\nabla F(x_t) - \nabla \widehat{F}_m(x_t) + \nabla \widehat{F}_m(x_t) - v_t) - 2 \cdot \frac{\gamma}{4}\|v_t\|_2^2 - \frac{\gamma(1 - S_F\gamma)}{2}\|v_t\|_2^2,$$

$$\le F(x_t) - \frac{\gamma(1 - S_F\gamma)}{2}\|v_t\|_2^2 + \gamma\Big(\|\nabla F(x_t) - \nabla \widehat{F}_m(x_t)\|_2^2 + \|\nabla \widehat{F}_m(x_t) - v_t\|_2^2\Big), \tag{35}$$

the last inequality comes from Young's inequality, more precisely, i.e.

$$-\gamma v_t^\top (\nabla F(x_t) - \nabla \widehat{F}_m(x_t)) - \frac{\gamma}{4}\|v_t\|_2^2 \le \gamma\|\nabla F(x_t) - \nabla \widehat{F}_m(x_t)\|_2^2. \tag{36}$$

$\square$

**Proposition B.3** *Denote*

$$\delta := \frac{L_g^2 S_f^2 \sigma_g^2}{m} + \frac{L_f^2 L_g^2}{N_1}$$

*for each iteration, we have*

$$F(x_{t+1}) \le F(x_t) - \frac{\gamma(1 - S_F\gamma)}{2}\|v_t\|_2^2 + \gamma\delta + \gamma^3 \sum_{i=(n_t-1)q}^{t-1} \frac{S_F^2}{N_2}\mathbb{E}\|v_i\|^2 \tag{37}$$

**Proof:** Apply Proposition B.1, B.2 and Lemma B.1, we have

$$F(x_{t+1}) \le F(x_t) - \frac{\gamma(1 - S_F\gamma)}{2}\|v_t\|_2^2 + \gamma\Big(\|\nabla F(x_t) - \nabla \widehat{F}_m(x_t)\|_2^2 + \|\nabla \widehat{F}_m(x_t) - v_t\|_2^2\Big)$$

$$\le F(x_t) - \frac{\gamma(1 - S_F\gamma)}{2}\|v_t\|_2^2 + \gamma\Big(\frac{L_f^2 \sigma_{gg}^2 + L_g^2 S_f^2 \sigma_g^2}{m} + \frac{L_f^2 L_g^2}{N_1} + \sum_{i=(n_t-1)q}^{t-1} \frac{S_F^2}{N_2}\mathbb{E}\|x_{i+1} - x_i\|_2^2\Big)$$

$$= F(x_t) - \frac{\gamma(1 - S_F\gamma)}{2}\|v_t\|_2^2 + \gamma^3 \sum_{i=(n_t-1)q}^{t-1} \frac{S_F^2}{N_2}\mathbb{E}\|v_i\|^2 + \gamma\delta. \tag{38}$$

which concludes the proof. $\qquad\square$

Now we proceed to the proof of the convergence rate of BSpiderBoost.

**Proof:** [Proof of Theorem 3.4] Based on Proposition B.3, we telescope in each epoch and take expectation,

$$
\mathbb{E}\big[F(x_{t+1})\big]
$$

$$
\leq \mathbb{E}\Big[F(x_{(n_t-1)q}) - \frac{\gamma(1-S_F\gamma)}{2}\sum_{j=(n_t-1)q}^{t}\|v_j\|_2^2 + \gamma^3\sum_{j=(n_t-1)q}^{t}\sum_{i=(n_t-1)q}^{j-1}\frac{S_F^2}{N_2}\mathbb{E}\|v_i\|^2 + \gamma\sum_{j=(n_t-1)q}^{t}\delta\Big]
$$

$$
\leq \mathbb{E}\Big[F(x_{(n_t-1)q}) - \frac{\gamma(1-S_F\gamma)}{2}\sum_{j=(n_t-1)q}^{t}\|v_j\|_2^2 + \gamma^3\sum_{j=(n_t-1)q}^{n_tq-1}\sum_{i=(n_t-1)q}^{t}\frac{S_F^2}{N_2}\mathbb{E}\|v_i\|^2 + \gamma\sum_{j=(n_t-1)q}^{t}\delta\Big]
$$

$$
= \mathbb{E}\Big[F(x_{(n_t-1)q}) - \frac{\gamma(1-S_F\gamma)}{2}\sum_{j=(n_t-1)q}^{t}\|v_j\|_2^2 + \frac{\gamma^3 S_F^2 q}{N_2}\sum_{i=(n_t-1)q}^{t}\mathbb{E}\|v_i\|^2 + \gamma\sum_{j=(n_t-1)q}^{t}\delta\Big]
$$

$$
\leq \mathbb{E}\Big[F(x_{(n_t-1)q}) - \sum_{i=(n_t-1)q}^{t}\Big(\beta\|v_i\|_2^2 - \gamma\delta\Big)\Big],
$$

$$\tag{39}$$

the second inequality follows from $n_tq > t$ and $j-1 < t$. Further telescoping for all iterations, we have

$$
\mathbb{E}\big[F(x_T) - F(x_0)\big] \leq \mathbb{E}\Big[-\sum_{i=0}^{T-1}\Big(\beta\|v_i\|_2^2 - \gamma\delta\Big)\Big] = \gamma T\delta - \beta\sum_{i=0}^{T-1}\mathbb{E}\|v_i\|_2^2. \tag{40}
$$

As a result, for $x_S$, its corresponding $v_S$ satisfies that

$$
\mathbb{E}\|v_S\|_2^2 \leq \frac{1}{T}\sum_{i=0}^{T-1}\mathbb{E}\|v_i\|_2^2 \leq \frac{F(x_0) - \mathbb{E}F(x_T)}{\beta T} + \frac{\gamma}{\beta}\delta \leq \frac{\Delta}{\beta T} + \frac{\gamma}{\beta}\delta. \tag{41}
$$

By substituting the parameter settings, we could further show that

$$
\mathbb{E}\Big[\frac{1}{3}\|\nabla F(x_S)\|_2^2\Big] \leq \mathbb{E}\Big[\|\nabla F(x_S) - \nabla\widehat{F}_m(x_S)\|_2^2 + \|\nabla\widehat{F}_m(x_S) - v_S\|_2^2 + \|v_S\|_2^2\Big]
$$

$$
\leq \frac{L_g^2 S_f^2 \sigma_g^2}{m} + \Big(\frac{L_f^2 L_g^2}{N_1} + \mathbb{E}\sum_{i=(n_S-1)q}^{S-1}\frac{S_F^2}{N_2}\mathbb{E}\|x_{i+1} - x_i\|_2^2\Big) + \Big(\frac{\Delta}{\beta T} + \frac{\gamma}{\beta}\delta\Big)
$$

$$
\leq \delta + \frac{\Delta}{\beta T} + \frac{\gamma}{\beta}\delta + \frac{S_F^2}{N_2}\mathbb{E}\sum_{i=(n_S-1)q}^{S-1}\mathbb{E}\|x_{i+1} - x_i\|_2^2
$$

$$
\leq \delta + \frac{\Delta}{\beta T} + \frac{\gamma}{\beta}\delta + \frac{\gamma^2 S_F^2}{N_2}\cdot\frac{q}{T}\sum_{i=0}^{T-1}\mathbb{E}\|v_i\|_2^2
$$

$$
\leq \delta + \frac{\Delta}{\beta T} + \frac{\gamma}{\beta}\delta + \frac{\gamma^2 S_F^2 q}{N_2}\Big(\frac{\Delta}{\beta T} + \frac{\gamma}{\beta}\delta\Big)
$$

$$
= \Big(1 + \frac{\gamma}{\beta} + \frac{\gamma^3 S_F^2 q}{N_2\beta}\Big)\delta + \frac{1}{\beta T}\Big(1 + \frac{\gamma^2 S_F^2 q}{N_2}\Big)\Delta
$$

$$
\leq \Big(1 + \frac{1}{2\beta S_F} + \frac{1}{16\beta S_F}\Big)\delta + \frac{1}{\beta T}\Big(1 + \frac{1}{8}\Big)\Delta
$$

$$\tag{42}$$

where the second inequality comes from Proposition B.1(d), Lemma B.1, and Equation (41); the third inequality follows from the fact that the probability such that $n_S = 1$ or $2, \cdots, n_T$ is less or equal to

$q/T$; recall the definition of $\delta$ and multiply both sides by 3, then we have

$$\mathbb{E}\|\nabla F(x_S)\|_2^2 \le \left(3 + \frac{3}{2\beta S_F} + \frac{3}{16\beta S_F}\right)\left(\frac{L_g^2 S_f^2 \sigma_g^2}{m} + \frac{L_f^2 L_g^2}{N_1}\right) + \frac{4\Delta}{\beta T} \tag{43}$$
$$\le \frac{\epsilon^2}{4} + \frac{\epsilon^2}{4} + \frac{\epsilon^2}{2} = \epsilon^2,$$

By Jensen's inequality for the function $x^2$, it holds that

$$\left(\mathbb{E}\|\nabla F(x_S)\|_2\right)^2 \le \mathbb{E}\|\nabla F(x_S)\|_2^2 \le \epsilon^2. \tag{44}$$

So $x_S$ is the stationary point we desire. The corresponding iteration complexity is

$$\lceil T/q \rceil N_1 + 2TN_2 = \mathcal{O}(\epsilon^{-3}), \tag{45}$$

and the sample complexity is

$$\lceil T/q \rceil N_1 m + 2TN_2 m = \mathcal{O}(\epsilon^{-5}). \tag{46}$$

which concludes the proof. $\qquad\square$

## C  Lower Bounds, Proof of Theorem 4.1

**Proof:**  We prove the lower bounds of $\Delta(\mathcal{A}, \mathcal{F}, \Phi)$ by constructing a hard instance of function and a hard instance of an oracle.

The hard instance $F(x)$ is such that $F(x)$ is a CSO objective and it satisfies that $x = (y, z)$ and $F(x) = F^c(y) + F^{sm}(z)$. It means that $F$ is separable on $y$ and $z$.

The oracle $\phi \in \Phi_m$ returns a biased function value and a gradient estimator of $F$. Since $F$ is separable, we specifically consider a hard oracle instance $\phi$ such that $\phi$ returns a biased function value estimator and a biased gradient estimator of $F^{sm}(z)$ due to the compositional structure on $z$ and it returns an unbiased function value estimator and an unbiased gradient estimator of $F^c(y)$.

Based on this specific hard instance construction on the function and the oracle, we could decompose the lower bounds into two parts: one on coordinate $y$ and the other on coordinate $z$. Note that the part on coordinate $y$ is the classical lower bounds using the stochastic first-order oracle. The part on coordinate $z$ is related to the extra bias term introduced by the biased oracle.

We first consider the (strongly) convex CSO function class. By Yao's principle [46], we have

$$\begin{aligned}
&\Delta_T^*(\mathcal{A}, \mathcal{F}, \Phi)\\
&= \inf_{A \in \mathcal{A}} \sup_{F \in \mathcal{F}} \sup_{\phi \in \Phi} \mathbb{E}\Delta_T^A(F, \phi, \mathcal{X})\\
&\ge \inf_{A \in \mathcal{A}^d} \mathbb{E}_{\{V,\phi\}} \Delta_T^A(F_V, \phi, \mathcal{X})\\
&= \inf_{A \in \mathcal{A}^d} \mathbb{E}_{\{V,\phi\}} \left[F_V(x_T^A(\phi)) - F_V(x_V^*)\right],\\
&\ge \inf_{A \in \mathcal{A}^d} \mathbb{E}_{\{V,\phi\}} \left[F_V^c(y_T^A(\phi)) - F_V^c(y_V^*) + F_V^{sm}(z_T^A(\phi)) - F_V^{sm}(z_V^*)\right]\\
&\ge \inf_{A \in \mathcal{A}^d} \mathbb{E}_{\{V,\phi\}} \left[F_V^c(y_T^A(\phi)) - F_V^c(y_V^*)\right] + \inf_{A \in \mathcal{A}^d} \mathbb{E}_{\{V,\phi\}} \left[F_V^{sm}(z_T^A(\phi)) - F_V^{sm}(z_V^*)\right]
\end{aligned} \tag{47}$$

where $\mathcal{A}^d$ represents the class of all deterministic algorithms, $x_T^A(\phi) = (y_T^A(\phi), z_T^A(\phi))$, $x_V^* = (y_V^*, z_V^*)$ is the minimizer of $F_V(x)$. We first consider the lower bounds on the $y$ part where the oracle returns an unbiased function value and gradient estimator. To lower bound $\inf_{A \in \mathcal{A}^d} \mathbb{E}_{\{V,\phi\}} \left[F_V^c(y_T^A(\phi)) - F_V^c(y_V^*)\right]$, the hard instance construction of $F_V^c(y)$ will satisfy the following conditions.

**Condition I**:

- For $V = +1$ or $V = -1$, if $yV \le 0$, then $F_V^c(y) \ge F_V^c(0)$.
- For $V = +1$ or $V = -1$, $F_V^c(0) - \inf_y F_V^c(y) = c_0$ where $c_0 \ge 0$ is a constant.

Suppose that Condition I holds, we have

$$
\begin{aligned}
&\inf_{A \in \mathcal{A}^d} \mathbb{E}_{\{V, \phi\}} \left[ F_V^{\mathrm{c}}(y_T^A(\phi)) - F_V^{\mathrm{c}}(y_V^*) \right] \\
&\geq \inf_{A \in \mathcal{A}^d} \mathbb{E}_{\{V, \phi\}} \left[ (F_V^{\mathrm{c}}(0) - F_V^{\mathrm{c}}(y_V^*)) \mathbb{I}\{y_T^A(\phi)V \leq 0\} \right] \\
&= \inf_{A \in \mathcal{A}^d} c_0 \mathbb{E}_{\{V, \phi\}} \left[ \mathbb{I}\{y_T^A(\phi)V \leq 0\} \right] \\
&= \inf_{A \in \mathcal{A}^d} c_0 \mathbb{P}(y_T^A(\phi)V \leq 0).
\end{aligned}
\tag{48}
$$

Notice that (48) requires to lower bound a probability

$$
\mathbb{P}_V\{y_T^A(\phi)V \leq 0\}.
$$

For a constant $v = +1$ or $v = -1$, let $\mathbb{P}^v$ denote the probability distribution of the following trajectory

$$
(y_0^A, G_v(y_0^A, \zeta_0), y_1^A(\phi), ..., G_v(y_{T-1}^A(\phi), \zeta_{T-1}), y_T^A(\phi)).
$$

It holds that

$$
\mathbb{P}_V\{y_T^A(\phi)V \leq 0\} \geq 1 - \|\mathbb{P}^{+1} - \mathbb{P}^{-1}\|_{\mathrm{TV}} \geq 1 - \sqrt{0.5 D_{\mathrm{KL}}(\mathbb{P}^{+1} \| \mathbb{P}^{-1})},
$$

where $\| \cdot \|_{\mathrm{TV}}$ denotes the total variation distance of two probability distributions, $D_{\mathrm{KL}}$ denotes the KL divergence of two probability distributions. The first inequality holds by definition and the second inequality comes from Pinsker's inequality [11]. Since $A \in \mathcal{A}^d$ is a deterministic algorithm, conditioned on oracle return $\phi(y_j, \zeta_j)$, $y_j^A(\phi)$ is deterministic. Conditioned on $y_j^A(\phi)$, the randomness in $\phi(y_j^A(\phi), \zeta_j)$ only comes from $\zeta_j$. By the chain rule of KL divergence, we have

$$
D_{\mathrm{KL}}\left(\mathbb{P}^{+1} \| \mathbb{P}^{-1}\right) = \sum_{t=0}^{T-1} D_{\mathrm{KL}}(G_{+1}(y_t^A(\phi_{+1}), \zeta_t)|y_t^A(\phi_{+1}) \| G_{-1}(y_t^A \phi_{-1}, \zeta_t)|y_t^A(\phi_{-1})).
$$

In our hard oracle construction, we require the following conditions:

**Condition II**

- The gradient estimator returned by the oracle, conditioned on the query point, is a normal random variable such that conditioned on $w_t$, it holds:

$$
G_V(y_t^A(\phi), \zeta_t))|y_t^A(\phi) \sim \mathbb{N}(\mu_V^A, \sigma^2)
$$

  where $\mu_V^A$ depends on the algorithm $A$, $\sigma^2$ is the variance parameter of the oracle.
- There exists a constant $b_y$ such that $|\mu_{+1}^A - \mu_{-1}^A| \leq b_y$.

Since the KL divergence between two normal random variables with the same variance $\sigma^2$ is known to be $\frac{(\mu_1 - \mu_2)^2}{2\sigma^2}$, where $\mu_1$ and $\mu_2$ are the expectations of the two normal random variables, respectively. As a result, we have

$$
D_{\mathrm{KL}}(\mathbb{P}^{+1} \| \mathbb{P}^{-1}) = \frac{T(\mu_{+1}^A - \mu_{-1}^A)^2}{2\sigma^2} \leq \frac{Tb_y^2}{2\sigma^2}.
$$

Thus, it implies that

$$
\mathbb{P}_V\{y_T^A(\phi)V \leq 0\} \geq 1 - \sqrt{\frac{T^2 b_y^2}{4\sigma^2}}.
\tag{49}
$$

Therefore we have the lower bounds on the first term on the right hand side of (47):

$$
\inf_{A \in \mathcal{A}^d} \mathbb{E}_{\{V, \phi\}} \left[ F_V^{\mathrm{c}}(y_T^A(\phi)) - F_V^{\mathrm{c}}(y_V^*) \right] \geq c_0 \left( 1 - \sqrt{\frac{T^2 b_y^2}{4\sigma^2}} \right).
\tag{50}
$$

Now we lower bound the second term $\inf_{A \in \mathcal{A}^d} \mathbb{E}_{\{V, \phi\}} \left[ F_V^{\mathrm{sm}}(z_T^A(\phi)) - F_V^{\mathrm{sm}}(z_V^*) \right]$ on the right hand side of (47). Let $\hat{F}_{m, V}^{\mathrm{sm}}(z)$ denote the expectation of the function value estimator returned by the oracle on $z$ part. Suppose the hard instance construction satisfies the following conditions.

**Condition III**:

- For $V = +1$ or $V = -1$, if $zV \leq 0$, then $F_V^{\mathrm{sm}}(z) \geq F_V^{\mathrm{sm}}(0)$.

- For $V = +1$ or $V = -1$, $F_V^{\mathrm{sm}}(0) - \hat{F}_{m,V}^{\mathrm{sm}}(z_V^*) = c_m$ where $c_m \geq 0$ for any $m \geq m_0$.

- For $V = +1$ or $V = -1$, $\hat{F}_{m,V}^{\mathrm{sm}}(z_V^*) - F_V^{\mathrm{sm}}(z_V^*) = c_m'$, where $c_m' \geq 0$ is a constant.

Condition III guarantees that for any $z$ such that $zV \leq 0$, it holds
$$F_V^{\mathrm{sm}}(z) - \hat{F}_{m,V}^{\mathrm{sm}}(z_V^*) \geq F_V^{\mathrm{sm}}(0) - \hat{F}_{m,V}^{\mathrm{sm}}(z_V^*) \geq c_m \geq 0.$$
We further have that
$$
\begin{aligned}
&\inf_{A \in \mathcal{A}^d} \mathbb{E}_{\{V,\phi\}} \left[ F_V^{\mathrm{sm}}(z_T^A(\phi)) - F_V^{\mathrm{sm}}(z_V^*) \right] \\
&\geq \inf_{A \in \mathcal{A}^d} \mathbb{E}_{\{V,\phi\}} (F_V^{\mathrm{sm}}(0) - F_V^{\mathrm{sm}}(z_V^*)) \mathbb{I}\{z_T^A(\phi)V \leq 0\} \\
&= \inf_{A \in \mathcal{A}^d} \mathbb{E}_{\{V,\phi\}} (F_V^{\mathrm{sm}}(0) - \hat{F}_{m,V}^{\mathrm{sm}}(z_V^*) + \hat{F}_{m,V}^{\mathrm{sm}}(z_V^*) - F_V^{\mathrm{sm}}(z_V^*)) \mathbb{I}\{z_T^A(\phi)V \leq 0\} \qquad (51) \\
&= \inf_{A \in \mathcal{A}^d} (c_m + c_m') \mathbb{P}\{z_T^A(\phi)V \leq 0\} \\
&\geq \inf_{A \in \mathcal{A}^d} c_m' \mathbb{P}\{z_T^A(\phi)V \leq 0\}.
\end{aligned}
$$
Similar as (49), we have
$$\mathbb{P}\{z_T^A(\phi)V \leq 0\} \geq 1 - \sqrt{\frac{T^2 b_z^2}{4\sigma^2}},$$
where $b_z \geq |\mu_1^A - \mu_{-1}^A|$ with $\mu_V^A$ as the expectation of the gradient estimator $G_V$ on $z$. To summarize, if Condition I, II, and III hold, we have
$$\Delta_T^*(\mathcal{A}, \mathcal{F}, \Phi) \geq c_0 \left( 1 - \sqrt{\frac{T^2 b_y^2}{4\sigma^2}} \right) + c_m' \left( 1 - \sqrt{\frac{T^2 b_z^2}{4\sigma^2}} \right). \qquad (52)$$

It remains to construct specific hard instances satisfying these conditions and demonstrate the lower bounds.

**Strongly convex CSO objective with smooth outer function**  The hard instance is
$$F_V(x) = \mathbb{E}_\xi[f_\xi(\mathbb{E}_{\eta|\xi} g_\eta(x, \xi))] = \frac{1}{2} y^2 - V\alpha y + \frac{1}{2} z^2 - V\beta z. \qquad (53)$$

Note that the outer function $f_\xi(y, z) = \frac{1}{2} y^2 - V\alpha y + \frac{1}{2} z^2 - V\beta z$ and the inner function $g_\eta(x, \xi) = (y, \eta z)$, where $\eta \sim \mathbb{N}(1, 1/\beta^2)$ for any $\xi$. The minimizer of $F_V(x)$ is $x_V^* = (V\alpha, V\beta)$. Further $F(x)$ can be decomposed as $F_V^c(y) = \frac{1}{2} y^2 - V\alpha y$ and $F_V^{\mathrm{sm}}(z) = \mathbb{E}_\xi[\frac{1}{2}(\mathbb{E}_{\eta|\xi}\eta z)^2 - V(\mathbb{E}_{\eta|\xi}\eta z\beta)] = \frac{1}{2} z^2 - V\beta z$. The oracle has access to an approximation function
$$\hat{F}_{m,V}(x) = F_V^c(y) + \hat{F}_{m,V}^{\mathrm{sm}}(z) = \frac{1}{2} y^2 - V\alpha y + \frac{1}{2} \mathbb{E}\hat{\eta}^2 z^2 - V\beta \mathbb{E}\hat{\eta} z = \frac{1}{2} y^2 - V\alpha y + \frac{1}{2}\left(1 + \frac{1}{m\beta^2}\right) z^2 - V\beta z,$$
where $\hat{\eta} = \frac{1}{m} \sum_{j=1}^m \eta_j$, and
$$F_V^{\mathrm{sm}}(z) = \frac{1}{2} z^2 - Vz\beta, \quad \hat{F}_V^{\mathrm{sm}}(z) = \frac{1}{2}\left(1 + \frac{\sigma_\eta^2}{m}\right) z^2 - Vz\beta.$$

. The gradient estimator returned by the oracle is
$$G_V(x) = (\nabla F^c(y), \nabla \hat{F}_{m,V}^{\mathrm{sm}}(z)) + \zeta),$$
where $\zeta \sim \mathbb{N}(0, \sigma^2)$. Thus $\mathbb{E}G_V(x) = (\nabla F^c(y), \nabla \hat{F}_{m,V}^{\mathrm{sm}}(z))) = \nabla \hat{F}_{m,V}(x)$. We verify that

- $F_V(x)$ is strongly convex and the outer function $f_\xi(y, z)$ is $\frac{1}{2}$-Lipschitz smooth.
- Condition I and III hold:
  - $c_0 = \frac{\alpha^2}{2}$.
  - $c_m = \frac{\beta^2}{2}(1 - \frac{1}{m\beta^2})$. Thus $c_m \geq 0$ if $m \geq 1/\beta^2$.

- $c'_m = \frac{1}{2m}$.
- Condition II holds:
    - $y - V\alpha + \zeta$ and $z - V\beta + \zeta$ are normal random variables conditioned on $V$ and $x$.
    - $b_y = 2\alpha$, $b_z = 2\beta$.

As a result we have

$$\Delta_T^*(\mathcal{A}, \mathcal{F}_{\mathrm{CSO}}^+, \Phi_{\mathrm{m}}) \geq \frac{\alpha^2}{2}(1 - \sqrt{\frac{T\alpha^2}{\sigma^2}}) + \frac{1}{2m}(1 - \sqrt{\frac{T\beta^2}{\sigma^2}}).$$

Thus there exists a hard instance with $\alpha = \frac{2}{3}\sqrt{\frac{\sigma^2}{T}}$, $\beta = \frac{1}{2}\sqrt{\frac{\sigma^2}{T}}$ such that

$$\Delta_T^*(\mathcal{A}, \mathcal{F}_{\mathrm{CSO}}^+, \Phi_{\mathrm{m}}) \geq \frac{4\sigma^2}{27T} + \frac{1}{4m}, \tag{54}$$

for strongly convex CSO objective with smooth outer function. As a result, to achieve $\epsilon$-optimality, the number of iterations should be at least $T = \mathcal{O}(\epsilon^{-1})$, the inner batch size should be at least $m = \mathcal{O}(\epsilon^{-1})$.

**Strongly convex CSO objective with non-smooth outer function**    We consider the hard instance construction:

$$F_V(x) = \mathbb{E}_\xi f_\xi(\mathbb{E}_{\eta|\xi} g_\eta(x, \xi)) = \frac{1}{2}y^2 - V\alpha y + \beta|z - V| + \beta^2(\frac{1}{2}z^2 - Vz),$$

with the outer function $f_\xi(y, z) = \frac{1}{2}y^2 - V\alpha y + \beta|z - V| + \beta^2(\frac{1}{2}z^2 - Vz)$ and the inner function $g_\eta(x, \xi) = (y, \eta z)$, where $\eta \sim \mathbb{N}(1, 1/\beta^2)$ for any $\xi$. The minimizer of $F_V(x)$ is $x_V^* = (V\alpha, V)$. Correspondingly, we have $F_V^c(y) = \frac{1}{2}y^2 - V\alpha y$ and $F_V^{\mathrm{sm}}(z) = \beta|z - V| + \beta^2(\frac{1}{2}z^2 - Vz)$. The oracle has access to an approximation function

$$\hat{F}_{m,V}(x) = F_V^c(y) + \hat{F}_{m,V}^{\mathrm{sm}}(z) = \frac{1}{2}y^2 - V\alpha y + \beta \mathbb{E}[|\hat{\eta}z - V| + \beta^2(\frac{1}{2}(\hat{\eta}z)^2 - V(\hat{\eta}z))].$$

The subgradient estimator returned by the oracle for $\hat{F}_{m,V}(x)$ is

$$G_V(x) = (\nabla F^c(y) + \zeta, \nabla \hat{F}_{m,V}^{\mathrm{sm}}(z) + \zeta)$$

where $\zeta \sim \mathbb{N}(0, \sigma^2)$. We verify that

- $F_V(x)$ is strongly convex and the outer function $f_\xi(y, z)$ is non-smooth.
- Condition I and III hold
    - $c_0 = \frac{\alpha^2}{2}$.
    - $c_m = \beta(1 - \mathbb{E}|\hat{\eta} - 1|) + \frac{\beta^2}{2}(1 - \mathbb{E}(\hat{\eta} - 1)^2)$. Since $\mathbb{E}|\hat{\eta} - 1| = \frac{1}{\beta\sqrt{m}}\sqrt{\frac{2}{\pi}}$, and $\mathbb{E}(\hat{\eta} - 1)^2 = \frac{1}{\beta^2 m}$, $c_m \geq 0$ if $1 \geq \frac{1}{\beta\sqrt{m}}\sqrt{\frac{2}{\pi}}$ and $1 \geq \frac{1}{\beta^2 m}$.
    - $c'_m = \frac{1}{\sqrt{m}}\sqrt{\frac{2}{\pi}} + \frac{1}{2m}$.
- Condition II is satisfied:
    - $\nabla F^c(y) + \zeta$ and $\nabla \hat{F}_{m,V}^{\mathrm{sm}}(z) + \zeta$ are normal random variables conditioned on $V$ and $x$.
    - $b_y = 2\alpha$, $b_z = 2\beta + 2\beta^2$ as $|\nabla \hat{F}_{m,-}^{\mathrm{sm}}(z) - \nabla \hat{F}_{m,+}^{\mathrm{sm}}(z)| \leq 2\beta + 2\beta^2$.

As a result, we have

$$\Delta_T^*(\mathcal{F}_{\mathrm{CSO}}^+, \Phi_{\mathrm{m}}, \mathcal{A}) \geq \frac{\alpha^2}{2}\left(1 - \sqrt{\frac{T\alpha^2}{\sigma^2}}\right) + \left(\frac{1}{\sqrt{m}}\sqrt{\frac{2}{\pi}} + \frac{1}{2m}\right)\left(1 - \sqrt{\frac{T(\beta + \beta^2)^2}{\sigma^2}}\right)$$

As a result, there exists a hard instance with $\alpha = \frac{2}{3}\sqrt{\frac{\sigma^2}{T}}$, $\beta + \beta^2 = \frac{1}{2}\sqrt{\frac{\sigma^2}{T}}$, such that

$$\Delta_T^*(\mathcal{A}, \mathcal{F}_{\mathrm{CSO}}^+, \Phi_{\mathrm{m}}) \geq \frac{4\sigma^2}{27T} + \frac{1}{2\sqrt{m}}\sqrt{\frac{2}{\pi}} + \frac{1}{4m}, \tag{55}$$

for strongly convex CSO objective with non-smooth outer function.

**Convex CSO objective with smooth outer function** We consider a hard instance that differs from the hard instance in the strongly convex objectives with a smooth outer function only in $F_V^c(y)$.

$$F_V(x) = \mathbb{E}_\xi[f_\xi(\mathbb{E}_{\eta|\xi}g_\eta(x,\xi))] = \alpha\mathbb{I}\{|y-V| > r\}(|y-V| - \frac{1}{2}r) + \alpha\mathbb{I}\{|y-V| \le r\}\frac{1}{2r}(y-V)^2 + \frac{1}{2}z^2 - Vz\beta.$$

where the inner function $g_\eta(x,\xi) = (y, \eta z)$, where $\eta \sim \mathbb{N}(1, 1/\beta^2)$ for any $\xi$ and the outer function

$$f_\xi(y,z) = \alpha\mathbb{I}\{|y-V| > r\}(|y-V| - \frac{1}{2}r) + \alpha\mathbb{I}\{|y-V| \le r\}\frac{1}{2r}(y-V)^2 + \frac{1}{2}z^2 - Vz\beta.$$

Thus $F_V(x)$ is a convex CSO objective with smooth outer function. Specifically, we have

$$F_V^c(y) = \alpha\mathbb{I}\{|y-V| > r\}(|y-V| - \frac{1}{2}r) + \alpha\mathbb{I}\{|y-V| \le r\}\frac{1}{2r}(y-V)^2,$$

where $r > 0$. We construct an oracle with gradient estimator such that:

$$G_V(x) = (\nabla F_V^c(y) + \zeta, \nabla \hat{F}_V^{\text{sm}}(z) + \zeta),$$

where $\zeta \sim \mathbb{N}(0, \zeta)$. We verify that

- $F_V(x)$ is convex and the outer function $f_\xi(y,z)$ is $\frac{1}{r}$-Lipschitz smooth.
- Condition I and III hold:
  - when $0 < r < 1$ and $m \ge 1/\beta^2$, we have $c_0 = \alpha(1 - r/2) \ge 0$, $c_m = \frac{\beta^2}{2}(1 - \frac{1}{\beta^2 m}) \ge 0$.
  - $c_m' = \frac{1}{2m}$.
- Condition II holds:
  - $\nabla F_V^c(y) + \zeta$ and $\nabla \hat{F}_V^{\text{sm}}(z) + \zeta$ are a normal random variables conditioned on $V$ and $x$.
  - $b_y = 2\alpha$, $b_z = 2\beta$.

As a result, we have,

$$\Delta_T^*(\mathcal{F}_{\text{CSO}}^0, \Phi_m, \mathcal{A}) \ge \frac{\alpha}{2}\left(1 - \sqrt{\frac{T\alpha^2}{\sigma^2}}\right) + \frac{1}{2m}\left(1 - \sqrt{\frac{T\beta^2}{\sigma^2}}\right). \tag{56}$$

Thus there exist a hard instance with $\alpha = \frac{1}{2}\sqrt{\frac{\sigma^2}{T}}$ and $\beta = \frac{1}{2}\sqrt{\frac{\sigma^2}{T}}$ such that

$$\Delta_T^*(\mathcal{F}_{\text{CSO}}^0, \Phi_m, \mathcal{A}) \ge \sqrt{\frac{\sigma^2}{64T}} + \frac{1}{4m} \tag{57}$$

for convex CSO objective with smooth outer function.

**Convex CSO objective with non-smooth outer function** We consider the hard instance construction such that
$$F_V(x) = \mathbb{E}_\xi f_\xi(\mathbb{E}_{\eta|\xi}g_\eta(x,\xi)) = \alpha|y-V| + \beta|z-V|.$$

where the inner function is $g_\eta(x,\xi) = (y, z - \eta)$ with $\eta \sim \mathbb{N}(0, 1/\beta^2)$ for any $\xi$ and the outer function $f_\xi(y,z) = \alpha|y-V| + \beta|z-V|$. The minimizer of $F_V$ is $x_V^* = (V, V)$. The oracle has access to an approximation function:

$$\hat{F}_{m,V}(x) = F_V^c(y) + \hat{F}_{m,V}^{\text{sm}}(z) = \alpha|y-V| + \beta\mathbb{E}|z-V-\hat{\eta}|.$$

The subgradient estimator return by the oracle for $\hat{F}_{m,V}(x)$ is

$$G_V(x) = (\nabla F^c(y) + \zeta, \nabla \hat{F}_{m,V}^{\text{sm}}(z) + \zeta)$$

where $\zeta \sim \mathbb{N}(0, \sigma^2)$ and we abuse the use of $\nabla$ to denote subgradient. We verify that

- $F_V(x)$ is convex and the outer function $f_\xi(y,z)$ is non-smooth.
- Condition I and III are satisfied by our construction:
  - $c_0 = \alpha$.

- $c_m = \beta(1 - \mathbb{E}|\hat{\eta}|)$, since $\mathbb{E}|\hat{\eta}| = \frac{1}{\beta\sqrt{m}}\sqrt{\frac{2}{\pi}}$, $c_m \geq 0$ if $1 \geq \frac{\sigma_\eta}{\sqrt{m}}\sqrt{\frac{2}{\pi}}$.

    - $c'_m = \beta\mathbb{E}|\hat{\eta}| = \frac{1}{\sqrt{m}}\sqrt{\frac{2}{\pi}}$.

- Condition II is satisfied:

    - $\nabla F_V^c(y) + \zeta$ and $\nabla\hat{F}_{m,V}^{\mathrm{sm}}(z) + \zeta$ are normal random variables conditioned on $V$ and $x$.
    - $b_y = 2\alpha$, $b_z = 2\beta$.

As a result, we have

$$\Delta_T^*(\mathcal{F}_{\mathrm{CSO}}^0, \Phi_\mathrm{m}, \mathcal{A}) \geq \alpha\left(1 - \sqrt{\frac{T\alpha^2}{\sigma^2}}\right) + \frac{1}{\sqrt{m}}\sqrt{\frac{2}{\pi}}\left(1 - \sqrt{\frac{T\beta^2}{\sigma^2}}\right).$$

Thus there exists a hard instance with $\alpha = \frac{1}{2}\sqrt{\frac{\sigma^2}{T}}$ and $\beta = \frac{1}{2}\sqrt{\frac{\sigma^2}{T}}$, such that

$$\Delta_T^*(\mathcal{A}, \mathcal{F}_{\mathrm{CSO}}^0, \Phi_\mathrm{m}) \geq \sqrt{\frac{\sigma^2}{64T}} + \frac{1}{2\sqrt{m}}\sqrt{\frac{2}{\pi}} \tag{58}$$

for convex CSO objective with non-smooth outer function.

Letting the right-hand side of each result be greater or equal to $\epsilon$, we have the corresponding sample complexity for each case.

For the nonconvex CSO problems, we construct a hard instance such that $F^c(y)$ is nonconvex smooth and $F^{\mathrm{sm}}(z)$ is 1-strongly convex with Lipschitz continuous or Lipschitz smooth outer function. Since $F^{\mathrm{sm}}(z)$ is 1-strongly convex, it holds

$$\|\nabla F^{\mathrm{sm}}(z)\|_2^2 \geq 2(F^{\mathrm{sm}}(z) - \inf_z F^{\mathrm{sm}}(z)).$$

We further have

$$\Delta_T^{*g}(\mathcal{A}, \mathcal{F}_{\mathrm{CSO}}^-, \Phi_m,)$$
$$= \inf_{A\in\mathcal{A}} \sup_{\phi\in\Phi_m} \sup_{F\in\mathcal{F}_{\mathrm{CSO}}^-} \mathbb{E}\|\nabla F(x_T^A(\phi))\|_2^2$$
$$\geq \inf_{A\in\mathcal{A}} \sup_{\phi\in\Phi} \sup_{F^c\in\mathcal{F}^-} \mathbb{E}\|\nabla F^c(y_T^A(\phi))\|_2^2 + \inf_{A\in\mathcal{A}} \sup_{\phi\in\Phi_m} \sup_{F^{sm}\in\mathcal{F}_{\mathrm{CSO}}^+} \mathbb{E}\|\nabla F^{\mathrm{sm}}(z_T^A(\phi))\|_2^2$$
$$\geq \sup_{\phi\in\Phi} \sup_{P_F\in\mathcal{P}\{\mathcal{F}^-\}} \inf_{A\in\mathcal{A}} \mathbb{E}\|\nabla F(y_T^A(\phi))\|_2^2 + \inf_{A\in\mathcal{A}} \sup_{\phi\in\Phi_m} \sup_{F^{sm}\in\mathcal{F}_{\mathrm{CSO}}^+} \mathbb{E}\|\nabla F^{\mathrm{sm}}(z_T^A(\phi))\|_2^2$$
$$\geq \sup_{\phi\in\Phi} \sup_{P_F\in\mathcal{P}\{\mathcal{F}^-\}} \inf_{A\in\mathcal{A}} \mathbb{E}\|\nabla F(y_T^A(\phi))\|_2^2 + \inf_{A\in\mathcal{A}} \sup_{\phi\in\Phi_m} \sup_{F^{sm}\in\mathcal{F}_{\mathrm{CSO}}^+} 2\mathbb{E}(F^{\mathrm{sm}}(z_T^A(\phi)) - \inf_z F^{\mathrm{sm}}(z)),$$

where $\Phi \subset \Phi_\mathrm{m}$ is the oracle class such that $\mathbb{E}h(x,\zeta) = F(x)$, the first inequality holds by definition, the second inequality uses the fact from Braun et al. [5] that

$$\inf_{A\in\mathcal{A}} \sup_{\phi\in\Phi} \sup_{F\in\mathcal{F}} \Delta_T^g(A, F, \phi) \geq \sup_{\phi\in\Phi} \sup_{F\in\mathcal{P}\{\mathcal{F}\}} \inf_{A\in\mathcal{A}} \Delta_T^g(A, F, \phi), \tag{59}$$

where $\mathcal{P}(\mathcal{F})$ is the set of all distributions over $\mathcal{F}$, the third inequality holds by strong convexity.

Note that the second term on the right hand side of the last inequality is exactly twice the minimax error for strongly convex CSO objectives. Thus we could use the hard instance construction earlier on the strongly convex $\hat{F}_{m,V}^{\mathrm{sm}}$ to lower bound it. When the oracle has access to an $\mathbb{E}\hat{F}(x; \xi, \{\eta_j\}_{j=1}^m)$ with smooth outer function $f_\xi$,

$$\inf_{A\in\mathcal{A}} \sup_{\phi\in\Phi_m} \sup_{F^{sm}\in\mathcal{F}_{\mathrm{CSO}}^+} 2\mathbb{E}(F^{\mathrm{sm}}(z_T^A(\phi)) - \inf_z F^{\mathrm{sm}}(z^*)) \geq \frac{1}{2m}.$$

When the oracle has access to an $\mathbb{E}\hat{F}(x; \xi, \{\eta_j\}_{j=1}^m)$ with nonsmooth outer function $f_\xi$,

$$\inf_{A\in\mathcal{A}} \sup_{\phi\in\Phi_m} \sup_{F^{sm}\in\mathcal{F}_{\mathrm{CSO}}^+} 2\mathbb{E}(F^{\mathrm{sm}}(z_T^A(\phi)) - \inf_z F^{\mathrm{sm}}(z)) \geq \frac{1}{2m} + \sqrt{\frac{2}{m\pi}}.$$

As for the first term on the right-hand side, we directly use the results from Arjevani et al. [3] to lower bound it. Arjevani et al. [3] says that for $\mathcal{F}_S^-$, the class of nonconvex smooth functions, it holds for any $\epsilon > 0$ that

$$\sup_{\phi \in \Phi} \sup_{P_F \in \mathcal{P}\{\mathcal{F}_S^-\}} \inf_{A \in \mathcal{A}} \mathbb{E}\|\nabla F\big(y_t^A(\phi)\big)\|_2^2 \geq \epsilon^2,$$

for any $t \leq t_{\max} = \mathcal{O}(\sigma^2 \epsilon^{-4})$ and

$$\sup_{\phi \in \Phi^c} \sup_{P_F \in \mathcal{P}\{\mathcal{F}_S^-\}} \inf_{A \in \mathcal{A}} \mathbb{E}\|\nabla F\big(y_t^A(\phi)\big)\|_2^2 \geq \epsilon^2,$$

for $t \leq t_{\max}^c = \mathcal{O}(\sigma^2 \epsilon^{-3})$ where $\Phi^c$ denote an stochastic first-order unbiased oracle class such that any oracle in this class has Lipschitz continuous gradient estimator. It implies that for any $T$, there exists an $\epsilon > 0$ such that $T = \epsilon^{-4} \leq t_{\max}$ and

$$\sup_{\phi \in \Phi} \sup_{P_F \in \mathcal{P}\{\mathcal{F}_S^-\}} \inf_{A \in \mathcal{A}} \mathbb{E}\|\nabla F\big(y_T^A(\phi)\big)\|_2^2 \geq \epsilon^2.$$

It further implies that

$$\sup_{\phi \in \Phi} \sup_{P_F \in \mathcal{P}\{\mathcal{F}^-\}} \inf_{A \in \mathcal{A}} \mathbb{E}\|\nabla F\big(y_T^A(\phi)\big)\|_2^2 \geq \sup_{\phi \in \Phi} \sup_{P_F \in \mathcal{P}\{\mathcal{F}_S^-\}} \inf_{A \in \mathcal{A}} \mathbb{E}\|\nabla F\big(y_T^A(\phi)\big)\|_2^2 \geq \mathcal{O}(\sigma T^{-1/2}),$$

and

$$\sup_{\phi \in \Phi^c} \sup_{P_F \in \mathcal{P}\{\mathcal{F}^-\}} \inf_{A \in \mathcal{A}} \mathbb{E}\|\nabla F\big(y_T^A(\phi)\big)\|_2^2 \geq \sup_{\phi \in \Phi^c} \sup_{P_F \in \mathcal{P}\{\mathcal{F}_S^-\}} \inf_{A \in \mathcal{A}} \mathbb{E}\|\nabla F\big(y_T^A(\phi)\big)\|_2^2 \geq \mathcal{O}(\sigma T^{-2/3}).$$

□

# D  Experiments

The platform used for the experiments is Intel Core i9-7940X CPU @ 3.10GHz, 32GB RAM, 64-bit Ubuntu 18.04.3 LTS.

## D.1  Invariant Logistic Regression

We generate a synthetic dataset with $d = 10$, $a \sim \mathcal{N}(0; \sigma_1^2 I_d)$ with $\sigma_1^2 = 1$ , $\eta|\xi \sim \mathcal{N}(a; \sigma_2^2 I_d)$. Three different variances of $\eta|\xi$: $\sigma_2^2 \in \{1, 10, 100\}$ are considered, corresponding to different noise ratios. At each iteration, we use a fixed mini-batch size $m_t = m$, namely $m$ samples of $\eta|\xi$ are generated for a given feature label pair $\xi_i = (a_i, b_i)$. We fine-tune the stepsizes for BSGD using grid search.

Table 4 compares the performance achieved by BSGD and SAA under the metric of optimality gap, $F(x) - F^*$. Since we do not have direct access to the function value. We estimate the objective with 50000 outer samples and calculate the true conditional expectation. The empirical risk minimization constructed by SAA is solved using CVXPY.

## D.2  MAML

We use the objective value as the measurement. Since the objective is analytically intractable, we evaluate the MAML objective via empirical objective obtained by empirical risk minimization:

$$\hat{F}(w) = \frac{1}{\hat{T}} \sum_{i=1}^{\hat{T}} \frac{1}{\hat{N}} \sum_{n=1}^{\hat{N}} l_i\Big(w - \alpha \cdot \frac{1}{\hat{M}} \sum_{m=1}^{\hat{M}} \nabla_w l_i(w, D_{support}^{i,m}); D_{query}^{i,n}\Big), \tag{60}$$

where the three sample sizes $\hat{T}$, $\hat{N}$ and $\hat{M}$ are set to be 100. when computing the approximate loss function value, the sample tasks/data are selected randomly.

Figure 3 shows that the widely used first-order MAML [19], which ignores the Hessian information when constructing the gradient estimator, may not converge. The number after each method denotes

Table 4: Comparison of BSGD and SAA in Invariant Logistic Regression

| | $\sigma_1 = 1, \sigma_2 = 1$ | | | | | |
| | $Q = 10^5$ | | $Q = 5 \times 10^5$ | | $Q = 10^6$ | |
| $m$ | Mean | Dev | Mean | Dev | Mean | Dev |
|---|---|---|---|---|---|---|
| 1 | **9.28e-04** | 1.95e-04 | 6.23e-04 | 8.18e-05 | 5.81e-04 | 4.00e-05 |
| 5 | 1.04e-03 | 3.06e-04 | **2.08e-04** | 6.54e-05 | **1.77e-04** | 4.70e-05 |
| 10 | 1.22e-03 | 2.15e-04 | 3.69e-04 | 8.14e-05 | 2.91e-04 | 4.91e-05 |
| 20 | 1.46e-03 | 8.94e-04 | 3.22e-04 | 1.54e-04 | 1.66e-04 | 6.44e-05 |
| 50 | 1.53e-02 | 3.47e-03 | 8.82e-04 | 3.56e-04 | 3.94e-04 | 1.61e-04 |
| 100 | 3.40e-02 | 8.58e-03 | 1.94e-03 | 6.48e-04 | 9.27e-04 | 3.45e-04 |
| **SAA (m=100)** | 2.55e-03 | 9.34e-04 | 8.95e-04 | 3.78e-04 | 5.56e-04 | 2.81e-04 |
| | $\sigma_1 = 1, \sigma_2 = 10$ | | | | | |
| | $Q = 10^5$ | | $Q = 5 \times 10^5$ | | $Q = 10^6$ | |
| $m$ | Mean | Dev | Mean | Dev | Mean | Dev |
| 1 | 2.47e-03 | 1.12e-03 | 1.02e-03 | 2.83e-04 | 8.16e-04 | 1.38e-04 |
| 5 | **2.21e-03** | 9.22e-04 | **5.53e-04** | 1.30e-04 | **3.26e-04** | 1.15e-04 |
| 10 | 2.32e-03 | 5.29e-04 | 7.22e-04 | 2.55e-04 | 5.32e-04 | 1.72e-04 |
| 20 | 3.57e-03 | 7.88e-04 | 7.37e-04 | 3.25e-04 | 3.99e-04 | 1.37e-04 |
| 50 | 7.87e-03 | 2.96e-03 | 1.42e-03 | 7.57e-04 | 7.25e-04 | 3.65e-04 |
| 100 | 1.91e-02 | 6.46e-03 | 2.23e-03 | 1.01e-03 | 8.90e-04 | 4.83e-04 |
| **SAA (m=464)** | 8.69e-03 | 2.74e-03 | 3.70e-03 | 1.07e-03 | 2.14e-03 | 8.45e-04 |
| | $\sigma_1 = 1, \sigma_2 = 100$ | | | | | |
| | $Q = 10^5$ | | $Q = 5 \times 10^5$ | | $Q = 10^6$ | |
| $m$ | Mean | Dev | Mean | Dev | Mean | Dev |
| 1 | 7.32e-02 | 7.94e-03 | 6.82e-02 | 2.41e-03 | 6.69e-02 | 1.09e-03 |
| 5 | 1.53e-02 | 4.54e-03 | 3.30e-03 | 1.12e-03 | 1.61e-03 | 7.50e-04 |
| 10 | **1.46e-02** | 3.80e-03 | 3.28e-03 | 1.24e-03 | 1.70e-03 | 5.82e-04 |
| 20 | 1.73e-02 | 8.95e-03 | **3.19e-03** | 1.18e-03 | 1.52e-03 | 5.60e-04 |
| 50 | 1.47e-02 | 5.15e-03 | 3.36e-03 | 1.27e-03 | **1.50e-03** | 6.97e-04 |
| 100 | 3.20e-02 | 8.07e-03 | 5.81e-03 | 2.44e-03 | 3.39e-03 | 1.30e-03 |
| **SAA (m=1000)** | 4.33e-02 | 1.19e-03 | 1.50e-02 | 8.00e-04 | 1.12e-02 | 6.42e-04 |

Figure 3: FO-MAML may not converge

the inner mini-batch size. It compares the convergences of the widely used First-order MAML(FO-MAML)[19], BSGD, and Adam, each under the best-tuned inner batch size. BSGD achieves the least error among the three methods with a proper inner batch size of 20. Adam requires a larger inner batch size to achieve its best performance, which is less practical as some tasks only have a few or even one sample.

Table 5 summarizes the detailed experimental results of BSGD, FO-MAML, Adam, and BSpiderBoost with different inner mini-batch sizes. The total sample size is $Q = 10^7$. The stepsizes for BSGD, FO-MAML, and BSpiderBoost are fine-tuned. Specifically for BSpiderBoost we set $(N_1 = 10, N_2 = 1, q = 10)$. For each inner mini-batch size, we run each algorithm for 10 times and then calculate the mean and the standard deviation of the output objectives of all trials. The best performance result for each algorithm is highlighted using bold font.

Table 5: Comparison of convergence results of BSGD, FO-MAML and Adam in MAML problem with different inner mini-batch sizes.

| Method | $m$ | $Q = 10^5$ | | $Q = 10^6$ | | $Q = 10^7$ | |
|---|---|---|---|---|---|---|---|
| | | *Mean* | *Dev* | *Mean* | *Dev* | *Mean* | *Dev* |
| **BSGD** | 5 | 3.46e+00 | 1.81e-01 | 1.28e+00 | 1.72e-01 | 5.07e-01 | 1.02e-01 |
| | 10 | **3.40e+00** | 2.43e-01 | **1.20e+00** | 2.68e-01 | 2.68e-01 | 8.09e-02 |
| | 20 | 3.57e+00 | 3.18e-01 | 1.67e+00 | 6.87e-01 | **1.63e-01** | 5.82e-02 |
| | 50 | 3.44e+00 | 2.07e-01 | 2.51e+00 | 6.12e-01 | 2.51e-01 | 5.21e-02 |
| | 100 | 3.81e+00 | 3.66e-01 | 3.23e+00 | 2.99e-01 | 3.60e-01 | 9.05e-02 |
| **FO-MAML** | 5 | 3.89e+00 | 3.46e-01 | 3.21e+00 | 2.49e-01 | 8.48e-01 | 1.59e-01 |
| | 10 | **3.72e+00** | 3.18e-01 | **2.07e+00** | 4.94e-01 | **2.58e-01** | 4.02e-02 |
| | 20 | 4.03e+00 | 3.32e-01 | 3.15e+00 | 1.69e-01 | 1.82e+00 | 7.12e-01 |
| | 50 | 3.90e+00 | 3.84e-01 | 3.26e+00 | 3.24e-01 | 3.52e+00 | 5.49e-01 |
| | 100 | 3.80e+00 | 3.62e-01 | 3.48e+00 | 2.51e-01 | 4.09e+00 | 5.10e-01 |
| **Adam** | 5 | **2.95e+00** | 5.90e-01 | 1.45e+00 | 4.15e-01 | 1.04e+00 | 3.83e-01 |
| | 10 | 3.03e+00 | 4.26e-01 | 1.34e+00 | 5.61e-01 | 6.09e-01 | 7.59e-01 |
| | 20 | 3.47e+00 | 3.01e-01 | **1.11e+00** | 3.63e-01 | 2.82e-01 | 8.85e-02 |
| | 50 | 3.36e+00 | 3.43e-01 | 1.53e+00 | 5.20e-01 | **2.35e-01** | 8.32e-02 |
| | 100 | 3.60e+00 | 2.86e-01 | 2.52e+00 | 5.28e-01 | 3.92e-01 | 2.20e-01 |
| **BSpiderBoost** | 10 | 2.47e+00 | 1.02e+00 | 2.65e+00 | 1.08e+00 | 2.56e+00 | 1.04e+00 |
| | 20 | 6.81e-01 | 6.50e-01 | **1.95e-01** | 4.49e-02 | **1.43e-01** | 2.57e-02 |
| | 50 | **6.15e-01** | 1.93e-01 | 2.54e-01 | 6.38e-02 | 2.10e-01 | 3.56e-02 |
| | 100 | 3.21e+00 | 1.12e+00 | 2.76e+00 | 1.45e+00 | 2.98e+00 | 1.55e+00 |

The header spanning row reads $\alpha = 0.01$.