[Reviews · NeurIPS 2020]

Review 1

Summary and Contributions: The paper focuses on Conditional Stochastic Optimization (CSO) setting, which aims the solve the constraint problems of the form \min_{x \in \mathcal X} F(x) := E_\xi f_\xi( E_{\eta \mid \xi} g_\eta(x, \xi) ). The setting has connections to the Stochastic Conditional Optimization (SCO); \min_{x \in \mathcal X} F(x) := E_\xi f_\xi( E_{\eta} g_\eta(x) ). Both CSO and SCO have recently attracted much attention due to their wide range of applications from reinforcement learning and meta-learning to finance. Authors propose a biased Stochastic Gradient Descent (BSGD) algorithm that constructs biased gradient estimates by drawing samples from P(\xi) and and P(\eta \mid \xi). Biasedness is a direct result of compositional structure of the problem and the dependence between inner and outer expectation due to conditional distribution. Also, authors solve the CSO problem directly without sample average approximation (SAA). Authors assume a setting where samples are available from both P(\xi) and P(\eta \mid \xi), which separates it from some of the prior work. The paper claims to present the first stochastic-gradient based algorithm for solving CSO and proves sample complexities for convex/strongly convex/weakly convex F and Lipschitz continuous/Lipschitz smooth f_\xi. Authors also prove lower bounds for sample complexity under these settings and argue about improvements over the prior work that applies to the CSO setting. They provide numerical results that demonstrates mini-batch/smoothness trade-off for BSGD, as well as performance with respect to total sample complexity in comparison to relevant algorithms under different applications. Note: I will defer particular questions and comments about the technical details, proofs and results to the section of Additional comments. Although that section is left for additional comments, I believe it is easier to address particular questions and comments all together.

Strengths: To the best of my knowledge, the BSGD algorithm is the first stochastic-gradient based algorithm that directly solves CSO problem itself. The two most relevant work that focus on CSO are [12] and [24]; [12] solves a saddle-point problem reformulation of CSO, while [24] resorts to providing sample complexities for SAA approach to solve general CSO problem. With respect to the SAA approach presented in [24], BSGD method improves in sample complexities (they remove the dependence on d) when F is general convex, matching the lower bounds they provide. Although BSGD is not optimal when F is strongly convex and smooth, it matches the complexities of SAA approach[24]. They also argue about the settings in which BSGD may not be optimal, providing a transparent evaluation of their algorithm. In addition to sample complexities for general F, authors argue that BSGD matches sample complexities of existing methods for MAML, and BSGD appears to be comparatively practical. Authors provide a lower bound analysis for convex/strongly convex and nonconvex (smooth) F, with first-order algorithms under biased oracles. The constructions for smooth and nonconvex F were based on prior work ([5] and[10]) and assume a particular algorithm class (zero-respecting algorithms) and bounded initialization. Although I am not an expert in lower bound analysis, I checked the proofs for the lower bound constructions and arguments, which were thorough and intricate. I believe it is especially important to evaluate sample complexities of algorithms for CSO with respect to lower bounds to have a concrete reference point for comparison. Existence of lower bound analysis is significant. Authors argue about possible applications where CSO problem arises and provide experiments under several of those scenarios. I would comment on experiments in details later, but authors successfully present connections to practical implications of the CSO setting and hence their algorithm. Due to relatively small inner sample complexity, BSGD has practical advantages.

Weaknesses: SCO has gained popularity recently and CSO is a particular variation to it, which has concrete implications in practice. Although authors discuss their approach in conjunction with the most relevant prior work, the related work section is rather short and not detailed enough. The authors could have discussed individual papers (or in groups based on their relevance to each other) to highlight the main differences and overlaps between settings/assumptions they have and the algorithmic frameworks. CSO setting itself has been motivated in introduction but the paper lacks the narrative that clarifies differences between relevant problems settings and algorithms that are tailored for those particular problems. I believe this would illustrate why and how the CSO setting is important in a broader perspective. Second, authors argue that dual embedding and the primal-dual approach in [12] does not apply to nonconvex objectives, however, there are settings where BSGD and the algorithm in [12] have an overlap (when problem is convex). As far as I could understand, authors could have discussed the sample complexities for such cases. Similarly, authors could have devised an experimental setup to compare both methods in practice. Overall, I do not have any major comments about the general weaknesses of the paper. I have a few questions for technical details, which I believe the Additional comments section is more suitable for. ==================== POST-REBUTTAL ==================== My major concern was on the related work section, and authors address it in their rebuttal. My other comments were mostly suggestions for clarifications. I believe the lower bound analysis, together with the sample complexity results makes a good paper for NeurIPS. I am increasing my score.

Correctness: I checked the proofs for sample complexities in full details and validated the consistency of lower bound analysis. I fervently believe that their sample complexity results, theorems in the main text, and the lower bound computations are correct. Worst-case sample complexity analysis is straightforward and they make use of classical approaches. Therefore, proofs steps seem to be correct. As for empirical approaches, authors provide the full description of the numerical setups in the main text and appendix. The paper is transparent with respect to how each experiment had been carried out. Experiments are devised in such a way that they provide observations about the algorithm’s behavior under different settings (smoothness of the outer function vs mini batch size trade-off) and it compares their method with respect to existing methods in a fair manner. I have a few constructive comments with respect to numerical procedures in the Additional comments section.

Clarity: I find the paper well-written; the structure is easy to follow and the authors make their claims and contributions clear throughout the main text and appendix. An overarching comment about the clarity is that authors sometimes did not define the random variable with respect to which the expectation is evaluated. Due to the compositional structure and the existence of conditional expectation, it is essential to make this clear; different definitions would imply different results. I will make detailed comments about this in the last section, although this appears to be a minor point. I appreciate the effort of the authors in explaining the intricate details and implications of their main results, as well as the limitation that their methodology possesses whenever necessary. Main results and contributions of the paper are highlighted and presented clearly.

Relation to Prior Work: Authors make their contributions clear and discuss how their results are different/better than the most relevant prior work (references [12] and [24]). As I discussed in the weaknesses, authors include a sufficient selection of related work in the paper, but they did not explain the differences between their method and the prior work, as well as differences among the relevant literature. Although there does not exist a large amount of prior work with respect to CSO setting, it is essential to discuss relationship to prior work in details.

Reproducibility: Yes

Additional Feedback: Comments and Questions for Related Work: - I would recommend the authors to dedicate a section in Related work to discussion of the literature on stochastic compositional optimization as this direction of research in stochastic optimization paved the way for the CSO. Discussion of differences between these settings and the respective applications would be beneficial for motivating the CSO setting. As I already discussed, the references in related work should be elaborated, at least briefly. Comments and Questions for Main Text: - I believe the sample complexities in [12] could have been analyzed under the settings where BSGD is also applicable. Would the authors consider to add these results, for instance to the Table 1? - For Assumption 1(b), I think the expectation should be conditioned on x, as it will be a random variable depending on history of \xi and \eta, and the towering property is invoked to compute the full expectation at some point. Please correct me if my statement is wrong. - Line 128-129. I think it would be important to specify the random variables with respect to which the expectations are evaluated, as it is important in compositional structure. Also, it would be better to discuss whether and how \xi and Y are dependent. - Variable mini-batch sizes ( for Lipschitz continuous/smooth f_\xi) for strongly-convex F was given wrong. Strongly-convex F; Lipschitz continuous f_\xi -> m_t = O(t^2), Lipschitz smooth f_\xi -> m_t = O(t). I later realized after checking the appendix that those quantities were correct there. Please check their definitions in the main text. - F_{\Delta, S} at line 219 changes to F_{S, \Delta} at line 227. Comments and Question for Appendix: - Line 436-437: I don't see the reason why E \zeta_{t2} = E[ F(x*) - F(x_t) ], when expectation was evaluated over { \xi, \{ \eta_{tj} \} }. Obviously, p(x*, \xi_t) and p(x_t, \xi_t) are independent of \eta_{tj} and in addition, E_xi_t p(x_t, \xi_t) = F(x_t) by definition. I assume expectation in E[ F(x*) - F(x_t) ] refers to full expectation, which is evaluated through towering property. Could the authors please clarify the computation at this line? The same computation occurs at lines 511-513, and this time there does not exist expectation around F. - As a follow up remark, - The last inequality in Eq. (22) does not require the use of Young's inequality. When the value of x_t - x_{t+1} is plugged in, the same bound is achieved with equality. - Eq after line 590: Last equality is obtained irrespective of ordering between \sigma_1^2 and \sigma_2^2. Could the authors briefly explain this step? - For MAML problem, could the authors explain why they haven't tuned Adam, please? Adam is sensitive to initial step size tuning, as most other stochastic optimization algorithms. In fact, a rough tuning of \beta values could make important changes in algorithms behavior. I would like to see how Adam would perform with a rought tuning of its hyper-parameters. It would be fair to compare algorithms under similar tuning strategies.


Review 2

Summary and Contributions: The paper describes an algorithm to solve "conditional" stochastic optimization problems. The main claims are around characterization of the rate of convergence of the presented algorithm (in terms of "sample complexity") under various conditions on the smoothness and lipschitz continuity of various functions involved.

Strengths: The optimization problem is interesting and worth studying for the NeurIPS community, especially as it pertains to MAML formulations. In addition, the authors bring up an important point that the degree of smoothness of the outer function in the composition can significantly alter the rate of convergence. However, significant problems exist in the discussion as I've outlined below.

Weaknesses: Post feedback: Thanks for your response, which does clarify some of the points I raised. I am happy to raise my score a notch. For assumption 1(b), I understand that this strong form lets you have the subsequent results, but given the problem setup it is better to start with assumptions on the functions f_i and g_i instead of their composition F. The dependence of "sample complexity" on m is still unclear: Specifically, suppose we take your first setup on line 149-150. Then, we take \eps^{-1} steps, where each minibatch is of size m_t=m=\eps^{-2}, so total computational effort expended is \eps^{-3} in order to get an optimality gap of \eps (from the first term of (5)). Isn't that inefficient compared to standard SGD on strongly convex problems, where \eps^{-1} steps of effort size O(1) leads to a gap of \eps ? There is an order O(\eps^{-2}) increase in effort in your case. I should have been more clear about what was bothering me about this in my original review, sorry about that. ------------ Several of the basic assumptions and derivations are puzzling and need stronger motivation and explanation. - Starting with the assumptions, why are the composite functions F being assumed to be strongly-convex etc.? Why not begin with assumptions on the building blocks g and f? Smoothness of various forms are assumed for f, but can you start with the more natural convexity assumptions on f and g instead of the current form? - What bearing does assumption 1(b) have on the bias of the gradient estimation? You are assuming that there is an upper bound on the second moment of the gradient estimate, which in effect bounds the sum of the variance and bias of the estimate. Shouldn't you be deriving these bounds rather than assuming them?

Correctness: There are several seeming errors that need to be cleared up: - The gradient formula for F in lines 101-102 need further elucidation. Is g going as R^d -> R^d and hence should have a Jacobian matrix instead of a gradient vector (usually denoted by \nabla)? Or is g going as R^d->R, in which case f is a function of a scalar and its derivative is more commonly written as df/dx ? Please explain. - Note that you have specifically assumed that the inner expectation is conditional on the outer random variable. So, assuming that in both outer and inner expectation and gradient operator can be interchanged, we get that: \nabla F = E _{\xi} [ \nabla f ( E _{\nu|\xi} g(x) )^T E _{\nu|\xi} \nabla g(x) ] In other words, you have a situation where \nabla F = E_{\xi} [ h_1 (x,\xi)^T h_2(x,\xi) ] where the two functions h_1 and h_2 are possibly highly correlated because of the common conditional expectation. How do you go from this to the expression you have in line 101-102 where the expectations are independent? - Additionally the role played by the sample size m in the reduction of bias in the various theorems are unclear. The size m is the only control available on the bias of the gradient, and so it is natural to assume that m must be made progressively larger to drive bias to zero if even convergence itself should be assured. Why do some of your results (e.g. Thm 2) solely drive step size to zero and claim convergence?

Clarity: The english is perfectly clear but the concept I had a hard time with.

Relation to Prior Work: there are an entire stream of work on estimating conditional expectations for stochastic optimization from the applied probability and mathematical finance community that the authors miss. This is a good representative ref: https://epubs.siam.org/doi/abs/10.1137/18M1173186

Reproducibility: Yes

Additional Feedback: n/a


Review 3

Summary and Contributions: This paper proposes a biased stochastic gradient descent (BSCD) algorithm for conditional stochastic optimization (CSO). The sample complexity of BSCD and the lower bound of sample complexity for stochastic oracle based algorithms are obtained in strongly convex, convex and weakly convex settings and when the outer function is Lipschitz continuous or Lipschitz smooth. The sample complexities of BSGD under general convex and smooth non-convex settings are tight. Numerical experiments demonstrate its superior performance over some existing algorithms for invariant logistic regression, model-agnostic meta-learning, and instrumental variable regression.

Strengths: (1) The CSO problem studied by this paper is significant since it covers important applications including MAML, planning and control in reinforcement learning, the optimal control in linearly-solvable Markov decision process, and instrumental variable regression in causal inference, etc. (2) The theoretical analysis is comprehensive as it covers a combination of 6 settings—whether the objective function is strongly convex, convex or weakly convex, and whether the outer function is Lipschitz continuous or Lipschitz smooth. (3) To my knowledge, this paper provides the first result of lower bound on sample complexity of stochastic oracle based algorithms for the CSO problem. (4) BSGD outperforms many relevant algorithms in the experiment.

Weaknesses: In “Yang, Shuoguang, Mengdi Wang, and Ethan X. Fang. "Multilevel stochastic gradient methods for nested composition optimization." SIAM Journal on Optimization 29.1 (2019): 616-659.” The authors propose the a-TSCGD algorithm that achieves sample complexity O(eps^{-4.5}) in the case of two-level composition (the CSO problem with 2 levels) in general non-convex and Lipschitz-smooth setting as shown in Theorem 3.2, which is lower than O(eps^{-6}) that achieved by BSGD. The sample complexity in convex and Lipschitz smooth setting is O(eps^{-4}) achieved by both papers. The sample complexity of a-TSCGD in strongly convex and Lipschitz smooth setting is O(eps^{-2.5}), which is worse than O(eps^{-2}) in this paper. Hence, the statement that “We propose the FIRST stochastic gradient-based algorithm for solving CSO problems and establish the FIRST sample complexity results of the algorithm under convex and nonconvex settings.” may not be true. I suggest the authors to clarify this and discuss their results and point out the technical difference. In table 1, the authors compare the sample complexities of BSGD and the SAA approach in [24]. However, in my understanding, they may have different meanings. The sample complexity of BSGD is the required number of samples for the BSGD algorithm to achieve a epsilon minimized of F, and the sample complexity of SAA is the required samples for the minimizer of the SAA problem to be an epsilon minimizer of F. I suggest the author to clarify and discuss this point. About reproducibility: Instrumental Variable Regression (IV): The approximate function h(w, X) is unknown.

Correctness: The algorithm is a correct generalization of SGD. The theorems are also well established. For the experiment implementation, I think it is valid if the same loss function and same approximator function is used among the algorithms in each experiment.

Clarity: Yes, I think the structure and sentences of this paper are very clear.

Relation to Prior Work: Yes, this paper clearly discussed its improvement from previous works.

Reproducibility: Yes

Additional Feedback: 2) There may be a typo in the final column of Table 1. Do you mean WC and Lipschitz? (3) The minimax error definition in Section 4.1 is inf_A sup_{phi} sup_F (error) while it is sup_{phi} sup_F inf_A … in Section 4.2. Why do you swap sup_{phi} sup_F and inf_A? (4) The caption of Figure 1: I think you mean $\sigma_a^2/\sigma_{\eta}^2$, yes? (5) I remember SAA is a finite-sum objective rather than an algorithm, yes? How do you obtain the solution to SAA in your experiment?


Review 4

Summary and Contributions: The authors study conditional stochastic optimization, i.e. min F(x) = E_xi f_xi(E_eta|xi f_eta(x, xi) For this problem, it is not possible to obtain unbiased estimation of the gradient nabla F. The authors propose to solve it using biased estimators of nabla F, sampling at every iteration one xi and m (eta_i) \sim eta|xi. Thanks to the law of large number, the bias is controlled by the batch-size. The proposed algorithm is then simply SGD, using this biased estimator. The authors adapt the proof of averaged SGD convergence (in the strongly-convex/convex/weakly-convex case), Lipschitz/smooth, and find that the average bias in the gradient estimator directly appear in the bound. Similarly, they adapt lower-bound estimation of SGD complexity under classical oracle settings, and find that the maximum bias directly appear as an additive term in the bounds. The authors assess the perfomance of their simple algorithm on three CSO tasks: synthetic invariant logistic regression, MAML on a simple task and instrumental variable regression on a toy dataset. ----------- EDIT AFTER REBUTTAL: My concerns about novelty and more general related work on SA (that would yield the present upper-bounds immediatly) have not been addressed.

Strengths: The paper is fairly well written (although some additional insights would be welcome, see below). It tackles an interesting problem, albeit a well studied one. Lower bounds are not surprising, but their derivations are highly technical (in particular, they required to understand existing technical proofs). They form the essential contribution of this work.

Weaknesses: - The upper bounds that the authors derive are not surprising, and requires little adaptation from textbook convergence proofs of SGD, where a bias term appears at each iteration. Theorem 4 is also a simple consequence of existing work on stochastic approximation with biased oracles, see for instance "Non-asymptotic Analysis of Biased Stochastic Approximation Scheme", Belhal Karimi et al, JMLR 2019, Theorem 1. Optimization with biased gradient is indeed a well studied problem, as that it appears in many other formulations than CSO. - Providing more information about the proof technique of the lower-bounds in the main text would be have been interesting. Some definitions made for Theorem 6 are hidden in the appendix, which makes the strength of Theorem 6 hard to assess. - As far as the reviewer understand, Theorem 5 is an adaptation of [23] and Theorem 6 an adaptation of [10]. The authors specifically say that their proof is an adaptation in the appendix (and in the main for [23]), but do not detail specifically where the adaptations are made to handle the bias, which makes it very hard to assess novelty and correctness.The first equality that changes in the proof of Theorem 5 is l. 594 p 19 of the appendix, which results in (31) then (32). The proof does not follow the same notation nor the same order as [23], but overall the bias term appears easy to handle. It is possible that the reviewer misses the difficulty, as it is not discussed. - The reviewer could only review superficially the proof of Theorem 6. It also appears to be a cosmetic adaptation of [10], and may be presented in a more concise way. - The experimental part is subpar, with synthetic tasks only. It allows to show the importance of the inner batch-size m, but the authors do not propose heuristic way of setting it. Biased SGD seems indeed a reasonable alternative to MAML for meta-learning, although a validation on real datasets would have been of great value.

Correctness: The proofs appears correct, although the reviewer may have missed some errors. The proofs should be rewritten to better underline novelty.

Clarity: The paper is well written overall. Some concepts are not well introduced: in particular, the notion of gradient mapping isn't introduced, and the notion of zero-respecting algorithm.

Relation to Prior Work: A substantial part of the literature on upper-bounds for optimisation with biased stochastic oracles appears to have been missed (see above)

Reproducibility: Yes

Additional Feedback: - The broader impact is too long and too far-fetched. This section should not be used for additional self-praise (the abstract is there for it). - WC Lipschitz upper-bound does not appear in Table 1, it would be nice to add it. - Are WC Lipchitz lower-bounds especially hard to get ? They do not appear in the manuscript. The authors could comment on this. - l 19: Should read when *f* is the identity function - It is a little confusing to see upper-bounds for non-smooth non-convex losses, and lower-bounds for smooth non-convex. The paper would be more readable by simplifying Theorem 4 and handle only the smooth case, with the non-smooth case moved to the appendix.

[Author Response · NeurIPS 2020]

We thank all reviewers for their detailed comments and acknowledgment of our contribution. We first address the
common question on the relation to prior work and then respond to each reviewer.

**Related Work:** Below we briefly point out the key differences from related work. As reviewers suggested, we will
dedicate a section for related work and discuss in greater details in our revision.

• **Prior work on SCO**: [37, 38] and [Yang et al, 2019] (mentioned by Reviewer 3) considered SCO problem:
$\min_{x \in \mathcal{X}} \mathbf{f} \circ \mathbf{g}(x) := \mathbb{E}_\xi \big[ f_\xi \big( \mathbb{E}_\eta [g_\eta(x)] \big) \big]$ and its $T$-level composition extension. They assumed that there exists a
*deterministic function* $\mathbf{g}(x)$, independent of $\xi$, such that $\mathbf{g}(x) := \mathbb{E}_\eta[g_\eta(x)|\xi]$. This does not hold for general CSO,
and their algorithms and analysis do not work for CSO.
• **Prior work on estimating conditional expectations:** [Giles et al., 2019 (mentioned by Reviewer 2), Hong and
Sandeep, 2009] focuses on pure estimation, whereas we deal with optimization, which is more challenging.
• **Prior work on biased gradient methods:** [23, Karimi et al., 2019 (mentioned by Reviewer 4), Hu et al., 2020,
Chen and Ronny, 2018]. These work focuses on general biased oracles, whereas in our work, the bias comes from
estimating the conditional expectation and is controllable by the inner sample size $m$ and the smoothness condition.

**Response to Reviewer 1:** Thanks for the careful reading. We will add the result in [12] in Table 1 and fix other issues.

**Response to Reviewer 2:** Thanks for the useful comments. In our humble opinion, *all issues brought up by the*
*reviewer are relatively minor and can be easily clarified.* We hope that our reply addresses the reviewer's concerns.

**Justification of assumptions:** We impose the assumption on $F$ to keep it general and will present conditions on $f$ and
$g$ such that the assumption is satisfied. For instance, $F$ is convex when (i) $f_\xi$ is convex and $g_\eta$ is linear; or (ii) $f_\xi$ and $g_\eta$
are convex and $f_\xi$ is non-decreasing. Strong convexity of $F$ holds when $l_2$-regularization is added to convex objectives.
Weak convexity of $F$ holds when (i) $f_\xi$ and $g_\eta$ are both Lipschitz continuous and smooth, or (ii) $f_\xi$ is convex and $g_\eta$ is
smooth [14]. Restricting to specific conditions on $f_\xi$ and $g_\eta$ may limit the generality of our results.

*Assumption 1(b)*: The reviewer is correct that the boundedness of gradient estimator can be derived (instead of assumed)
under certain conditions. For instance, (i) when $f_\xi$ and $g_\eta$ are Lipschitz continuous, Assumption 1(b) holds with
$\beta = 0$, $M^2 = L_f^2 L_g^2$; (ii) when $F$ and $\hat{F}$ are $\mu$-strongly convex and $S$-smooth, $\mathcal{X} = \mathbb{R}^d$, Assumption 1(b) holds
with $\beta = 2S^2/\mu^2$, $M^2 = 2\mathbb{E}\|\nabla \hat{F}(x^*)\|_2^2$ [Nguyen et al., 2018]. This is because $\mathbb{E}\|\nabla \hat{F}(x)\|_2^2 \le 2\mathbb{E}\|\nabla \hat{F}(x) -$
$\nabla \hat{F}(x^*)\|_2^2 + 2\mathbb{E}\|\nabla \hat{F}(x^*)\|_2^2 \le 2S^2\|x - x^*\|_2^2 + 2\mathbb{E}\|\nabla \hat{F}(x^*)\|_2^2 \le 2S^2/\mu^2\|\nabla F(x)\|_2^2 + 2\mathbb{E}\|\nabla \hat{F}(x^*)\|_2^2$. where the
second inequality modifies Thm 1 in [Johnson and Zhang, 2013]. However, beyond these situations, the boundedness of
gradient estimator may not automatically hold. We will clarify this assumption in the paper.

**Notation on derivative and typo in Line 101:** We will clarify that $\nabla$ is abusively used to denote the Jacobian
matrix, gradient vector, and derivative for simplicity. We apologize for the typo in Line 101. The correct equation is
$\nabla F(x) = \mathbb{E}_\xi \big[ (\mathbb{E}_{\eta|\xi} \nabla g_\eta(x,\xi))^\top \nabla f_\xi(\mathbb{E}_{\eta|\xi} g_\eta(x,\xi)) \big]$. Both issues do not affect our analysis and contribution.

**Effects of $m$ in Theorems:** We explicitly define the bias term $\Delta_f(m)$ in Eq. (3) between Lines 130 and 131 and use
it in Theorems 2, 3, and 4 to show the dependence of the upper bounds on the inner sample size $m$. The bias term is
$\mathcal{O}(1/m)$ when $f_\xi$ is Lipschitz smooth and $\mathcal{O}(1/\sqrt{m})$ if $f_\xi$ is Lipschitz continuous.

**Response to Reviewer 3:** CSO is not a special case of [Yang et al., 2019] when $T = 2$. Please refer to the above
discussion on Related Work. We compare with SAA [24] in Table 1 as both results of BSGD and SAA characterize
the sample complexity of finding an $\epsilon$-optimal solution of the original objective $F$. However, they require different
computation complexities since SAA requires solving the empirical problem.

**Additional comments:** (2) As marked in the footnote of Table 1, there is a difference in the lower bound when $\hat{F}$ is
Lipschitz continuous or Lipschitz smooth. (3) Both lower bound definitions are valid (see [31] for discussion on their
relationship); note that the first definition fixes the iteration $T$ while the second fixes the inaccuracy $\epsilon$. (5): The solution
to SAA is obtained by CVXPY solver.

**Response to Reviewer 4:** We will discuss the technical novelty of our lower bound proof and fix other issues.
**Results not surprising:** While the algorithmic idea of BSGD and analysis seem intuitive, these have never been studied
before in the context of CSO problems, which have various important applications in emerging fields like MAML. We
provide a comprehensive analysis of stochastic first-order methods for CSO in convex and nonconvex regimes, with
both upper and lower bounds. We do find several results quite interesting (to some extent, surprising):

• The smoothness of the outer function plays an important role in the total sample complexity of CSO.
• Simple algorithm such as the BSGD already attains near-optimal complexities for solving CSO, which is quite
different from recent results on nested stochastic optimization (requires sophisticated algorithms for different settings).
• Our lower bounds in nonconvex setting build on a novel construction of a worst-case biased gradient estimator by
perturbing the last coordinate; this may look straightforward in hindsight. Nonetheless, these results seem new.
• Albeit its simplicity, our numerical study shows that BSGD performs quite well on MAML tasks for meta-learning.

[Meta-Review · NeurIPS 2020]

During the rebuttal phase the reviewers did not come to a consensus. Two reviewers--especially R4, believe that the paper should not get accepted since the authors base their analysis on well known ideas. The other two think that the results are novel enough and are of the interest of the NeurIPS community. I tend to agree with the latter, so I will therefor accept.